# Exploratory phosphoproteomics profiling of *Aedes aegypti* Malpighian tubules during blood meal processing reveals dramatic transition in function

Yashoda Kandel[1], Matthew Pinch[1], Mahesh Lamsal[1], Nathan Martinez[1], Immo A. Hansen[1,2]*

1 Department of Biology, New Mexico State University, Las Cruces, New Mexico, United States of America, 2 Institute of Applied Biosciences, New Mexico State University, Las Cruces, New Mexico, United States of America

* immoh@nmsu.edu

## Abstract

Malpighian tubules, the renal organs of mosquitoes, facilitate the rapid dehydration of blood meals through aquaporin-mediated osmosis. We performed phosphoproteomics analysis of three Malpighian tubule protein-libraries (1000 tubules/sample) from unfed female mosquitoes as well as one and 24 hours after a blood meal. We identified 4663 putative phosphorylation sites in 1955 different proteins. Our exploratory dataset reveals blood meal-induced changes in phosphorylation patterns in many subunits of V-ATPase, proteins of the target of rapamycin signaling pathway, vesicle-mediated protein transport proteins, proteins involved in monocarboxylate transport, and aquaporins. Our phosphoproteomics data suggest the involvement of a variety of new pathways including nutrient-signaling, membrane protein shuttling, and paracellular water flow in the regulation of urine excretion. Our results support a model in which aquaporin channels translocate from intracellular vesicles to the cell membrane of stellate cells and the brush border membrane of principal cells upon blood feeding.

## Introduction

Female *Aedes aegypti* are anautogenous, meaning they require proteins and other nutrients from vertebrate blood for egg production [1, 2]. When these mosquitoes take a blood meal, they typically ingest up to twice their own body weight in blood, which impairs their mobility [3, 4]. Additionally, the blood meal alters homeostasis of salt and water in their circulatory fluid called hemolymph. If not properly controlled, this ion and water imbalance can place extreme osmotic stress on tissues throughout the whole mosquito [5]. As a result, rapid activation of a robust and homeostatic system is needed to maintain appropriate water, ion, and nutrient levels after blood ingestion.

During blood-feeding mosquitoes start urine excretion to eliminate excess water and sodium from their bodies [6]. In fact, the first drops of urine are usually excreted by

**Data Availability Statement:** All relevant data are within the manuscript and its Supporting information files.

**Funding:** IAH received funding through grant # 5SC1GM125584 from the National Institutes of Health (https://www.nih.gov/). IAH also received funding through grant #R35GM144049 from the National Institutes of Health (https://www.nih.gov/). The funders had no role in study design, data collection and analysis, decision to publish, or preparation of the manuscript.

**Competing interests:** The authors have declared that no competing interests exist.

mosquitoes while they are still in the process of taking a blood meal. This rapid post-blood meal (PBM) excretion can be classified into three phases [7, 8]. The initial peak phase occurs within the first 20–35 minutes PBM, and urine flow can reach rates greater than 40 nL/min. During this phase, much of the water and $Na^+$ from the plasma fraction of the blood meal are eliminated. The post peak phase which follows sees a decline in urine flow rate and can last up to 70 minutes PBM. The final phase, referred to as the late phase, can last through 2–3 hours PBM, and is defined as the time in which urine flow is only slightly elevated above the pre-blood meal baseline. The extreme increase in urine flow allows up to 40% of the blood meal volume to be excreted within the first few hours after engorgement [8–11].

Malpighian Tubules (MTs) are the primary excretory tissue in almost all insects including *Ae. aegypti*, and they are functionally analogous to vertebrate kidneys [12, 13]. MTs in insects are clusters of tubular organs that open into the insect hindgut. The number of tubules varies across different insect taxa, but in *Ae. aegypti*, this excretory organ consists of a cluster of five tubules. Each tubule is composed of two major cell types, principal cells (PC) and stellate cells (SC). Together, both cell types facilitate the net flux of ions and water into the lumen of the tubule for primary urine production [14, 15]. PCs primarily perform the active transport of sodium and potassium ions whereas SCs are smaller, comprise only 20% of the MT surface area, and are involved in the regulated flow of water and chloride ions. Ions, waste molecules, and water enter the lumen in the distal and medial segments of MTs, and solutes and ions are reabsorbed in the proximal ends, the hindgut, and the rectum [12]. Any remaining water and waste are finally excreted as urine.

According to the model proposed by Beyenbach and coworkers in 1987, the process of insect diuresis starts with the release of the neurohormone, calcitonin-like diuretic hormone, also referred to as diuretic hormone 31 (DH31), which stimulates cells of the MTs to produce cAMP [8, 16–19]. cAMP is a second messenger molecule that stimulates cation import into PCs across their basolateral membranes, creating an osmotic gradient between MT cells and the lumen of the tubule [7]. In particular, $Na^+$ import is stimulated by cAMP-mediated activation of $Na^+$ channels and $Na^+/K^+/2Cl^-$ co-transporters, which both removes excess $Na^+$ from the hemolymph, and allows for $Na^+$ to serve as a driver of osmotic water flux [7]. In PCs, Vacuolar-type ATPase pumps (V-ATPase) hydrolyze ATP and pump protons into the MT lumen [7, 12, 13]. This proton gradient is then used to power the transcellular secretion of $Na^+$ and $K^+$ ions into the MT lumen [7, 12, 13]. Negatively charged $Cl^-$ ions follow into the MT lumen to maintain electroneutrality [7, 12, 13]. These processes generate the osmotic gradient that powers the mass flow of water into the MT lumens both by paracellular movement and transcellular transport through aquaporins (AQPs) in SCs [7, 12, 13, 20–22]. Aquaporins (AQPs) are six transmembrane-domain proteins that facilitate the transport of water and small solutes across the cell membrane and are responsible for a large portion of the water excretion mediated by MTs [10, 23–25].

Along with transcellular flow, water and ions may also enter the MT lumen by paracellular flow. Insects have cell-cell adhesion structures analogous to tight junctions and adherens junctions [20, 26, 27]. These junctions consist of transmembrane proteins that physically link to other proteins on opposing cells, the actin cytoskeleton, and anchor proteins that join the transmembrane proteins to actin filaments. Tight junction analogues in insects are called septate junctions, and they span a wider area near the apical ends of epithelial cells compared to tight junctions [20, 26, 27]. However, they also allow for tight control of water and solute flow [20]. Modifications of cell-cell junction proteins have previously been implicated to provide an alternative path for $Cl^-$ and water flux across epithelial borders in mosquito MTs via paracellular flow [20].

Previous studies have shown that the post-translational regulation of ion transporters and AQPs happen via two different mechanisms: gating, and trafficking [28, 29]. Gating, which is the activation and deactivation of a membrane channel to regulate the movement of molecules is a form of post-translational modification that allows a quick response to sudden environmental stress (Fig 1). It is the quickest mechanism of membrane transporter activation as the transporters are already inserted in the plasma membrane and only need to be modified in order to be activated. This mechanism is common in plants and yeast [28, 29]. The exact structural detail about how the gating happens can be found in previous studies [30]. Regulation of insect AQP activation is not well studied, and while there is evidence for gating of several mammalian AQPs, many questions remain about the importance of this means of activation [28, 31]. In addition to directly regulating AQP activity, phosphorylation has also been implicated in altering accessibility of other regulatory proteins to binding sites in membrane bound AQPs, and to regulate trafficking of AQPs to and from the plasma membrane [28, 31]. Thus, while there is some evidence of phosphorylation directly gating AQP activity in mammalian cells, much evidence points to phosphorylation indirectly affecting AQP activity through other effector proteins or by trafficking [28, 30–33].

The second mechanism of eukaryotic membrane transporter regulation is trafficking. In general, trafficking means the movement of molecules within the cells. Membrane transporter trafficking occurs when transporters are stored in intracellular vesicles and transported to the plasma membrane (Fig 1). Subsequently, the transporters can be endocytosed, and recycled back to storage vesicles via recycling endosomes. Regulation of transporter activity by trafficking has been well-studied in several mammalian transporters including the mammalian GLUT4 glucose transporter [34–36] and AQP2 in human and rat kidneys [37]. Many proteins are known to be involved in trafficking of membrane transporters; for example, Rab proteins which act as membrane organizers, SNAREs that mediate vesicle fusion, F-actin, motor proteins, ubiquitin, and small GTPases [38, 39].

In addition to gating and trafficking existing proteins to regulate excretion, MT cells may also synthesize more proteins to meet the demands placed on them or to replace damaged existing transporters (Fig 1).

Which of these three regulatory mechanisms are used to regulate urine production in the Malpighian tubules of mosquitoes is unclear. The three are not necessarily exclusive. Since mosquitoes start the process of urine production immediately upon starting a blood meal [8, 9], expression of new AQPs and other transport proteins in MTs to activate diuresis does not seem likely. However, previous transcriptomic studies showed that AQP expression was increased in MTs of adult female *Ae. aegypti* at 3 hours PBM [10, 25]. Hence our hypothesis was that rapid aquaporin and solute transporter activation observed in *Ae. aegypti* MTs PBM occurs via post-translational modification rather than through new gene expression.

The goal of this study was to improve our understanding of the mechanisms that regulate ions and water balance in mosquitoes after a blood meal. We investigated changes in MT protein phosphorylation by pooling MTs from 200 mosquitoes to generate single samples of 1000 tubules each, at three time points before and after blood feeding. The three time points we chose for our phosphoproteomics analysis cover the MTs in three very different states. The initial time point included unfed mosquito females that were approximately 72 hours after hatching and in a reproductive state-of-arrest [40]. Little diuresis is happening during this stage [41]. The second time point, one hour after a blood meal, covers the MT at their peak of activity during the rapid dehydration of the blood meal [6]. At the third time point 24 hours PBM, the initial dehydration of the blood meal is completed but digestion of the blood proteins, hemoglobin and serum proteins, is in full swing producing toxic heme and a large suite of other metabolic products [42, 43]. We identified a large number of phosphorylation targets in

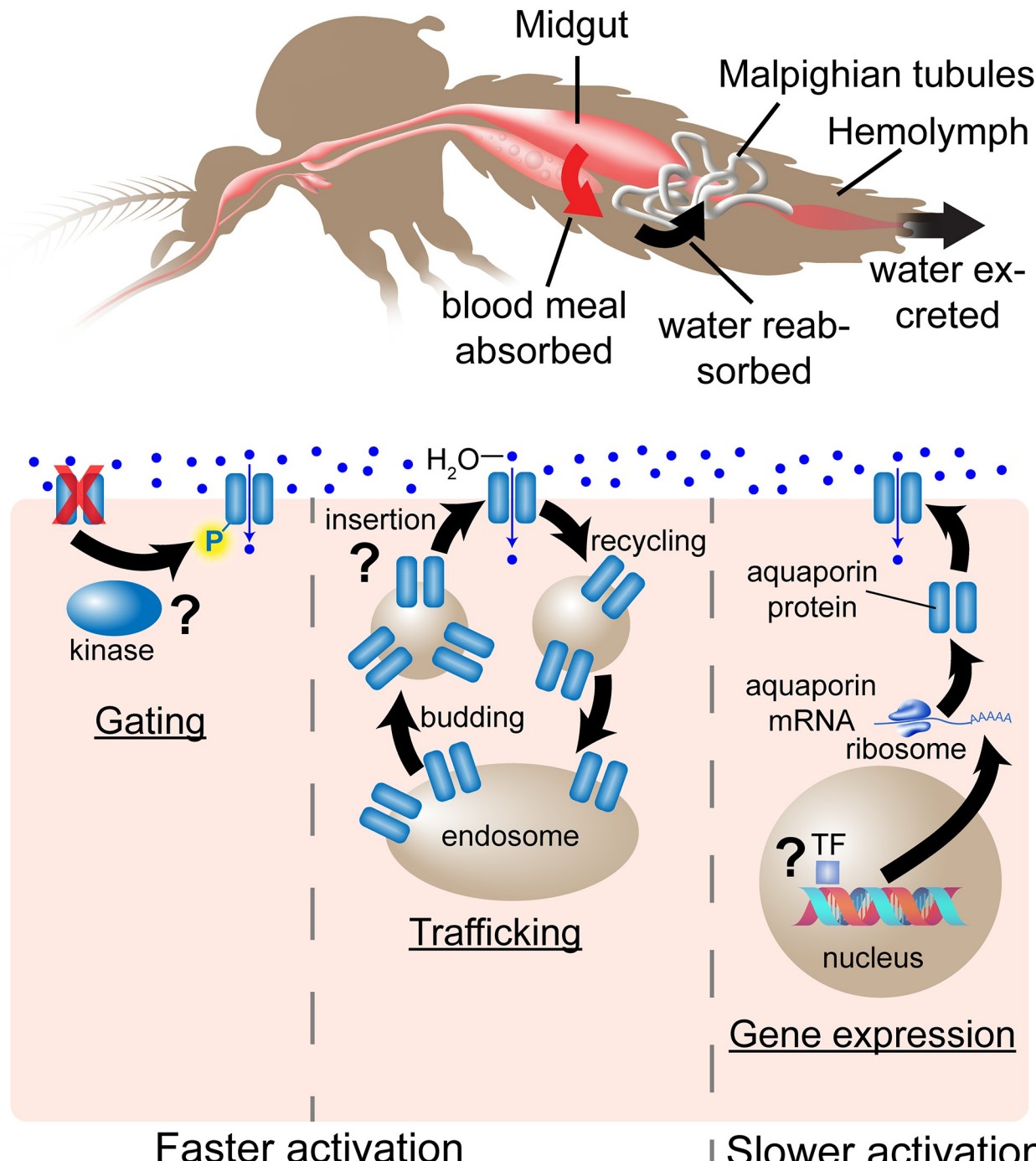

**Fig 1. Alternative hypothetical mechanisms of transporter activation in MT cells, using AQPs as an example.** Top: After blood feeding, female mosquitoes must rapidly eliminate large amounts of water and sodium from the ingested blood meal. Water, ions and nutrients from the blood meal are absorbed from the midgut into the hemolymph, and excess water and solutes are re-absorbed by the MTs and excreted in the process called diuresis. Bottom: Illustrated are three possible mechanisms of transporter activation in MT cells during diuresis. The quickest mechanism of activation is gating, or activation of transporters already inserted into the plasma membrane by post-translational modification such as phosphorylation or membrane voltage changes. A second rapid method of activation is trafficking of storage vesicles containing existing transporters to the plasma membrane. The slowest of the three mechanisms is de novo gene expression of transporter proteins in response to diuresis signaling.

proteins associated with signaling, ion transport, and vesicle-associated transport. The results of our exploratory study provide a unique opportunity for the development of hypotheses to identify novel targets for mosquito control strategies and show the value of *Ae. aegypti* as a model system for understanding the mechanisms of water and ion homeostasis in blood-feeding arthropods.

## Materials & methods

### Mosquito rearing and sampling

*Ae. aegypti* Liverpool strain mosquitoes were used in all experiments. After being laid, eggs were desiccated for at least one week. Eggs were then hatched in 13" x 20" pans at 27˚C in deionized (DI) water. Every three days, larvae were fed with dry cat food pellets (Special Kitty, Walmart Stores inc., Bentonville, AR). Every fifth day, the water in the pans was changed. Pupae were transferred to hatching dishes containing fresh DI water and dishes were placed into BugDorm-1 insect rearing cages (30x30x30 cm; BugDorm, Taichung, Taiwan) and allowed to emerge under controlled conditions (27˚C, 80% humidity, 14:10 h light:dark cycle). Prior to blood-feeding post-eclosion (PE) adults were fed on 20% sucrose solutions *ad libitum*.

One week PE, adult *Ae. aegypti* females were fed on defibrinated bovine blood (HemoStat Laboratories, Dixon, CA) for five min. (1 hr PBM samples) or 30 min. (24 hr PBM samples). One hr and 24hr post blood meal mosquitoes were anesthetized on ice and Malpighian Tubules (MT) from 200 mosquitoes (total of 1000 tubules per sample) were dissected in modified *Aedes* physiological saline (mAPS) solution from each group. Additionally, MTs were dissected from 200 unfed mosquitoes as a control. All dissected MTs from each time point were pooled to generate single samples, which were frozen in 200μL mAPS containing 2μL Halt™ protease inhibitor cocktail (Thermo Scientific, Rockford, IL) and 2μL Halt™ phosphatase inhibitor cocktail (Thermo Scientific, Rockford, IL). Frozen samples were shipped on dry ice to Creative Proteomics (Shirley, New York), who performed protein extractions, phospho-peptide enrichment, and phosphoproteomics analysis.

### Protein extraction

Samples were stored at -80˚C prior to analysis. To extract total protein, samples were thawed on ice and 4 volumes of lysis buffer (8 M urea, 1% protease inhibitor, 1% phosphatase inhibitor) were added. The lysed sample was sonicated and centrifuged at 12,000 xg for 10 min at 4˚C. Supernatant was transferred to a new centrifuge tube and protein concentration was determined by using a BCA kit.

### Protein digestion

Protein solution was transferred into Microcon devices YM-10 (Millipore, Burlington, MA) and centrifuged at 12,000g at 4˚C for 10 min. Subsequently, 200 μL of 50 mM ammonium bicarbonate was added to the concentrate followed by centrifugation and the process was repeated once. 10 mM DTT was added at 56˚C for 1 h to reduce disulfide bonds, and cystine residues were alkylated by 20 mM IAA at room temperature in the dark for 1h. The device was centrifuged at 12,000g at 4˚C for 10 min and wash once with 50 mM ammonium bicarbonate. 100 μL of 50 mM ammonium bicarbonate and free trypsin were added into the protein solution at a ratio of 1:50, and the solution was incubated at 37˚C overnight. Finally, the device was centrifuged at 12,000g at 4˚C for 10 min. 100 μL of 50 mM ammonium bicarbonate was added into the device and centrifuged, and then the process was repeated once again.

## Phosphopeptide enrichment

Fe-IMAC beads were resuspended by end-over-end rotation for 10 min. The beads were washed with 1 mL wash buffer three times. The peptide solution was transferred into the vial containing beads, and the whole solution was incubated on a rotator for 30 min at room temperature at maximum speed. The beads were allowed to settle and then the supernatant was removed taking care not to remove any beads. The beads were washed by adding 1 mL wash buffer, mixed by inverting the tube until the beads were completely resuspended. The beads were allowed to settle, the supernatant was removed. The washing step was repeated two more times and remaining supernatant was removed. 50 uL of elution buffer was added to the beads, and the beads were resuspended by tapping the bottom of the tube. The beads were allowed to re-settle. A new tube was rinsed with 0.5 mL ACN to remove any contaminants, vortexed, and discarded. The supernatant, containing the eluted phospho-peptides was transferred off the beads to the new ACN-rinsed tube. 40 μL of 20% TFA was added to acidify the eluate. The elution step was repeated and the elution fractions were combined to generate final eluted samples. The eluted phospho-peptides were dried in a speed-Vac and resuspended with 40 μL of 0.1% FA.

## Nano LC-MS/MS analysis

Phospho-peptides were separated by liquid chromatography using an Ultimate 3000 nano UHPLC system (ThermoFisher Scientific, USA). One μg of each sample was loaded into a Pep-Map C18, 100Å, 100 μm ×2 cm, 5 μm trapping column followed by a PepMap C18, 100Å, 75 μm ×50 cm, 2 μm analytical column. Samples were separated in a two-solvent linear gradient (solvent A: 0.1% formic acid in water; solvent B: 0.1% formic acid in 80% acetonitrile) with a total flow rate of 250 nL/min. as follows: first gradient from 2 to 8% solvent B in 3 min, then from 8% to 20% solvent B in 60 min, next from 20% to 40% solvent B in 23 min, then finally from 40% to 90% solvent B in 4 min.

The full MS scan was performed between 300–1,650 m/z with a resolution of 60,000 at 200 m/z. The automatic gain control target for the full scan was set to 3e6. The MS/MS scan was operated in Top 20 mode using a resolution of 15,000 at 200 m/z. The automatic gain control target was set to 1e5, normalized collision energy set at 28%, and an isolation window of 1.4 Th was used. Charge sate exclusion was as follows: unassigned, 1, > 6 with dynamic exclusion 30 s.

## Data analysis

Raw MS files were analyzed and searched against the *Aedes aegypti* protein database using MaxQuant (1.6.2.14). Sample protein data were normalized using the MaxLFQ package [44] in MaxQuant. Protein modifications included in the analysis were carbamidomethylation (C), oxidation (M) (variable), Phospho (STY) (variable). Additionally, enzyme specificity was set to trypsin, the maximum missed cleavages were set to 2, the precursor ion mass tolerance was set to 10 ppm, and MS/MS tolerance was 0.6 Da. A cut-off of 1.5-fold increased detection of phosphorylated residues was used to compile sets of proteins with differential phosphorylation between unfed and 1hr or 24hr PBM time points.

Datasets for each time point were manually curated to validate UniProt gene identities using Entrez Gene and VectorBase databases. Groups of proteins associated with particular physiological processes or signaling pathways were manually identified and retained for analysis.

## Results

### Phosphoprotein identification

We identified a total of 1955 phosphorylated proteins (including 4663 unique phosphorylation sites). The number of differentially phosphorylated proteins (DPPs) is summarized in Table 1. Detailed phosphoprotein and peptide identification information are listed in S1 File. Phosphoproteins presented in figures and in the text are highlighted in S2 File, and Phosphoproteins with greater than 25-fold increase in at least one PBM sample relative to the unfed sample are presented in S3 File.

### Vacuolar proton pump and ion exchangers

*Ae. aegypti* MTs consist of two cell types: large principal cells (PCs) and small stellate cells (SCs) (Fig 2a). Previous studies have shown that the active transport of $Na^+$ and $K^+$ ion into the lumen of MT from principal cells is powered by a V-type ATPase (Fig 2b) [7, 12, 13, 45–47]. The cytoplasmic complex of V-type ATPase hydrolyzes ATP, and membrane-bound complex pumps $H^+$ into the lumen of the MT [7, 12, 13, 48]. Protons secreted into the lumen of MT are exchanged for $Na^+$ and $K^+$ by antiporters (Fig 2b). Seven unique phosphorylation sites were identified within four of the fourteen V-ATPase subunits (Fig 2c and 2d). All seven did not show a change in phosphorylation at 1 hour PBM but all seven showed decreased phosphorylation at the 24 hours PBM time point.

Similarly, there was increased phosphorylation of sodium/hydrogen exchanger 3 (NHE3) 1hr PBM. A S770 site in this ion exchanger had a 187-fold increased phosphorylation 1hr PBM (Fig 2c). Phosphorylation of sodium-chloride cotransporters, and chloride ion channels also changed within 1hr PBM (Fig 2c).

### Aquaporins

Interestingly, phosphorylation of aquaporins did not change between unfed and 1hr PBM, except for one serine residue (S246) on aquaporin AQPAe.a (DRIP) which had lower phosphorylation levels at both time points PBM relative to unfed mosquitoes (Fig 2c). At 24hrs PBM all AQPs appear dephosphorylated (Fig 2c).

### Proteins involved in the endomembrane/secretory pathway

We observed changes in phosphorylation of a large number of proteins involved in intracellular vesicle transport systems (Figs 3 **and** 4). These changes include altered phosphorylation of co-translational machinery including subunits of both the signal recognition particle (SRP72) and translocation machinery necessary for insertion of membrane proteins (Sec61b and TRAM1), as well as proteins involved in ERES structure and COPII-coated vesicle budding

**Table 1. Comparison of differentially phosphorylated proteins before and after the blood meal.**

| Sample Name | Up-regulated (FC>1.5) | Down-regulated (FC<1/1.5) |
|---|---|---|
| Unfed Vs 1H PBM | 596 (432) | 524 (379) |
| Unfed VS 24H PBM | 794 (516) | 842 (561) |

The numbers outside the parenthesis represents number of phosphorylation sites and the number inside the parenthesis represents number of unique phosphorylated proteins. Fold-change (FC) > 1.5 was considered to be up-regulation, and FC < 0.67 was considered to be down-regulation in each PBM time point relative to unfed mosquitoes.

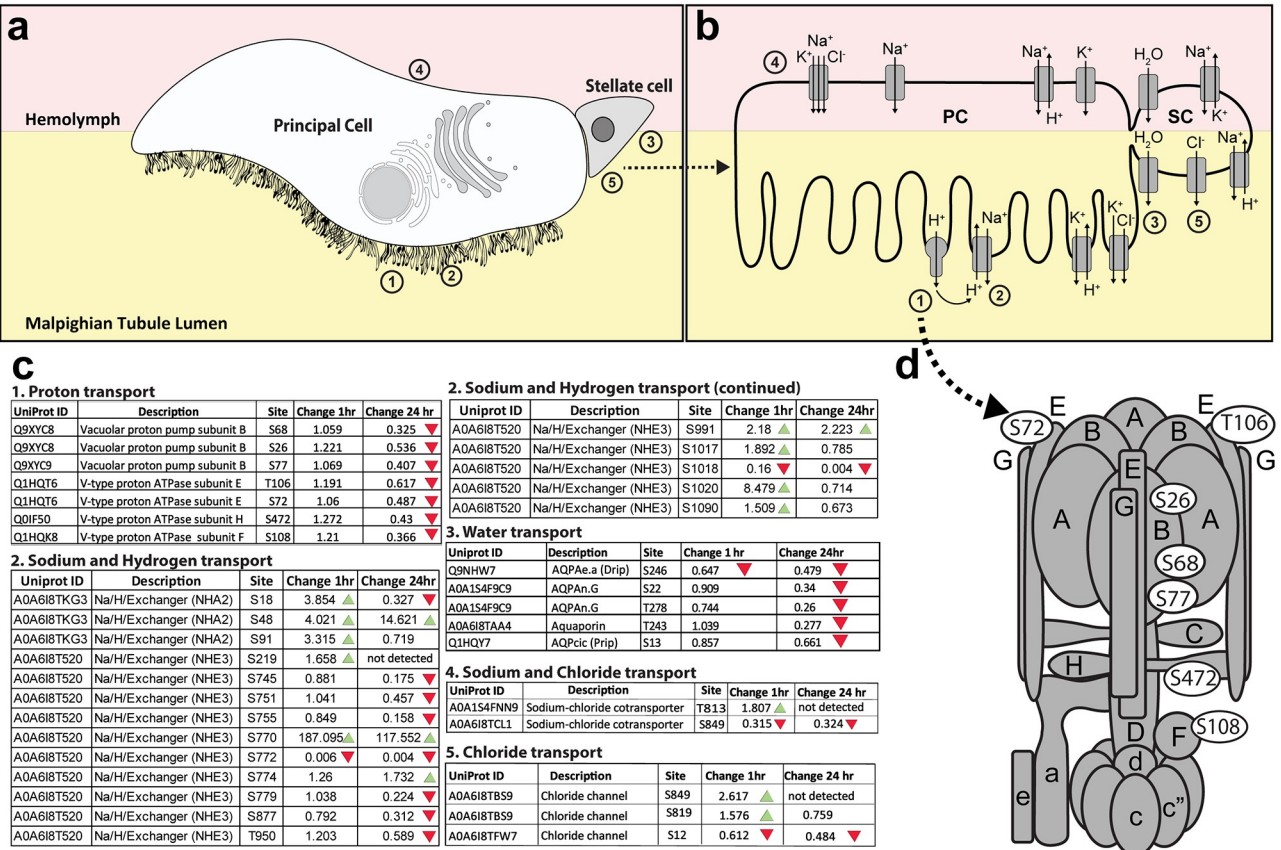

**1. Proton transport**

| UniProt ID | Description | Site | Change 1hr | Change 24hr |
|---|---|---|---|---|
| Q9XYC8 | Vacuolar proton pump subunit B | S68 | 1.059 | 0.325 ▼ |
| Q9XYC8 | Vacuolar proton pump subunit B | S26 | 1.221 | 0.536 ▼ |
| Q9XYC9 | Vacuolar proton pump subunit B | S77 | 1.069 | 0.407 ▼ |
| Q1HQT6 | V-type proton ATPase subunit E | T106 | 1.191 | 0.617 ▼ |
| Q1HQT6 | V-type proton ATPase subunit E | S72 | 1.06 | 0.487 ▼ |
| Q0IF50 | V-type proton ATPase subunit H | S472 | 1.272 | 0.43 ▼ |
| Q1HQK8 | V-type proton ATPase subunit F | S108 | 1.21 | 0.366 ▼ |

**2. Sodium and Hydrogen transport**

| Uniprot ID | Description | Site | Change 1hr | Change 24hr |
|---|---|---|---|---|
| A0A6I8TKG3 | Na/H Exchanger (NHA2) | S18 | 3.854 ▲ | 0.327 ▼ |
| A0A6I8TKG3 | Na/H Exchanger (NHA2) | S48 | 4.021 ▲ | 14.621 ▲ |
| A0A6I8TKG3 | Na/H Exchanger (NHA2) | S91 | 3.315 ▲ | 0.719 |
| A0A6I8T520 | Na/H Exchanger (NHE3) | S219 | 1.658 ▲ | not detected |
| A0A6I8T520 | Na/H Exchanger (NHE3) | S745 | 0.881 | 0.175 ▼ |
| A0A6I8T520 | Na/H Exchanger (NHE3) | S751 | 1.041 | 0.457 ▼ |
| A0A6I8T520 | Na/H Exchanger (NHE3) | S755 | 0.849 | 0.158 ▼ |
| A0A6I8T520 | Na/H Exchanger (NHE3) | S770 | 187.095 ▲ | 117.552 ▲ |
| A0A6I8T520 | Na/H Exchanger (NHE3) | S772 | 0.006 ▼ | 0.004 ▼ |
| A0A6I8T520 | Na/H Exchanger (NHE3) | S774 | 1.26 | 1.732 ▲ |
| A0A6I8T520 | Na/H Exchanger (NHE3) | S779 | 1.038 | 0.224 ▼ |
| A0A6I8T520 | Na/H Exchanger (NHE3) | S877 | 0.792 | 0.312 ▼ |
| A0A6I8T520 | Na/H Exchanger (NHE3) | T950 | 1.203 | 0.589 ▼ |

**2. Sodium and Hydrogen transport (continued)**

| Uniprot ID | Description | Site | Change 1hr | Change 24hr |
|---|---|---|---|---|
| A0A6I8T520 | Na/H Exchanger (NHE3) | S991 | 2.18 ▲ | 2.223 ▲ |
| A0A6I8T520 | Na/H Exchanger (NHE3) | S1017 | 1.892 ▲ | 0.785 |
| A0A6I8T520 | Na/H Exchanger (NHE3) | S1018 | 0.16 ▼ | 0.004 ▼ |
| A0A6I8T520 | Na/H Exchanger (NHE3) | S1020 | 8.479 ▲ | 0.714 |
| A0A6I8T520 | Na/H Exchanger (NHE3) | S1090 | 1.509 ▲ | 0.673 |

**3. Water transport**

| Uniprot ID | Description | Site | Change 1 hr | Change 24hr |
|---|---|---|---|---|
| Q9NHW7 | AQPAe.a (Drip) | S246 | 0.647 ▼ | 0.479 ▼ |
| A0A1S4F9C9 | AQPAn.G | S22 | 0.909 | 0.34 ▼ |
| A0A1S4F9C9 | AQPAn.G | T278 | 0.744 | 0.26 ▼ |
| A0A6I8TAA4 | Aquaporin | T243 | 1.039 | 0.277 ▼ |
| Q1HQY7 | AQPcic (Prip) | S13 | 0.857 | 0.661 ▼ |

**4. Sodium and Chloride transport**

| UniProt ID | Description | Site | Change 1hr | Change 24 hr |
|---|---|---|---|---|
| A0A1S4FNN9 | Sodium-chloride cotransporter | T813 | 1.807 ▲ | not detected |
| A0A6I8TCL1 | Sodium-chloride cotransporter | S849 | 0.315 ▼ | 0.324 ▼ |

**5. Chloride transport**

| UniProt ID | Description | Site | Change 1hr | Change 24hr |
|---|---|---|---|---|
| A0A6I8TBS9 | Chloride channel | S849 | 2.617 ▲ | not detected |
| A0A6I8TBS9 | Chloride channel | S819 | 1.576 ▲ | 0.759 |
| A0A6I8TFW7 | Chloride channel | S12 | 0.612 ▼ | 0.484 ▼ |

**Fig 2. Changes in phosphorylation of ion channels, pumps, and aquaporins. a.** Illustration of the two major cell types in the Malpighian tubule epithelium: principal cells (PCs) are larger and have an apical brush border of microvilli filled with mitochondria containing V-type H⁺ ATPase pumps and other ion transporters; stellate cells (SCs) are smaller and participate in Cl⁻ and water transport. Numbers correspond to different regions where particular transport mechanisms occur, and correspond to the numbers in parts b and c. **b.** Ion channels and aquaporins associated with transcellular transport are illustrated. **c.** Specific amino acid residues on proteins with phosphorylation changes of at least 1.5-fold between unfed and blood-fed mosquitoes are shown in white ovals. The numbers in the figure correspond to tables where proteins with related functions are grouped. Tables contain the Uniprot IDs for each protein as well as the fold-change in phosphorylation of each residue between unfed and either 1hr PBM or 24hr PBM mosquitoes. Green arrows represent an increase of at least 1.5-fold in phosphorylation at the specified residue between unfed and PBM mosquitoes, and red arrows represent a decrease of at least 1.5-fold in phosphorylation at the specified residue between unfed and PBM mosquitoes. Numbers of each table correspond to transport mechanisms illustrated in parts a and b. **d.** V-type H⁺ ATPase model adapted from **Zhao et al (2015)** and **Beyenbach & Piermarini (2011)** showing the location of subunits with differential phosphorylation. Subunits are shown in gray, and specific phosphorylated residues are denoted with white ovals.

(Fig 3). Of particular interest is the threonine residue (Thr498) of the COPII complex protein, Sec31 which had a 998-fold increase in phosphorylation 1hr PBM relative to unfed mosquitoes. We also identified changes in phosphorylation of proteins involved in ensuring these COPII-coated vesicles localize properly to the cis-Golgi, including Rab1A, and proteins involved in localizing and tethering vesicles to the cis-Golgi (TRAPPC10, TRAPPC11, and GolgA2). Recycling of components from the cis-Golgi back to the ER, and between Golgi stacks is also vital for the maintenance of anterograde transport through the endomembrane system. Therefore, we specifically searched our dataset for proteins involved in retrograde transport via COPI-coated vesicles. We identified several proteins associated either with COPI-coated vesicle budding from the cis-Golgi (Gbf1) or fusion of vesicles with the ER membrane (Use1 and Sec20) whose phosphorylation state changed post-blood feeding.

**1. Translation at the ER membrane**

| UniProt ID | Description | Site | change 1hr | | change 24hr | |
|---|---|---|---|---|---|---|
| A0A6I8T7W0 | SRP72 | S544 | 0.854 | ▼ | 0.368 | |
| A0A6I8T7W0 | SRP72 | T547 | 1.64 | ▲ | 0.502 | ▼ |
| A0A6I8T7W0 | SRP72 | S629 | 7.794 | ▲ | | |
| Q1HR43 | Sec61β | S55 | 1.748 | ▲ | 6.199 | ▲ |
| Q1HRQ8 | TRAM1 | S363 | 7.852 | ▲ | 4.82 | ▲ |

**4. COPI coat complex formation**

| UniProt ID | Description | Site | change 1hr | | change 24hr | |
|---|---|---|---|---|---|---|
| A0A1S4FN24 | Gbf1 | T257 | 1.568 | ▲ | 2.809 | ▲ |
| A0A1S4FN24 | Gbf1 | S1257 | 0.207 | ▼ | 0.452 | ▼ |
| A0A1S4FN24 | Gbf1 | S1262 | 0.884 | | 2.04 | ▲ |
| A0A1S4FN24 | Gbf1 | S1276 | 1.019 | | 2 | ▲ |

**5. Retrograde transport**

| UniProt ID | Description | Site | change 1hr | | change 24hr | |
|---|---|---|---|---|---|---|
| A0A1S4FDE9 | Sec20 | S145 | 1.527 | ▲ | 1.741 | ▲ |
| A0A1S4FDE9 | Sec20 | S158 | 1.948 | ▲ | 1.409 | |
| A0A6I8TER6 | Use1 | S75 | 1.52 | ▲ | 1.249 | |

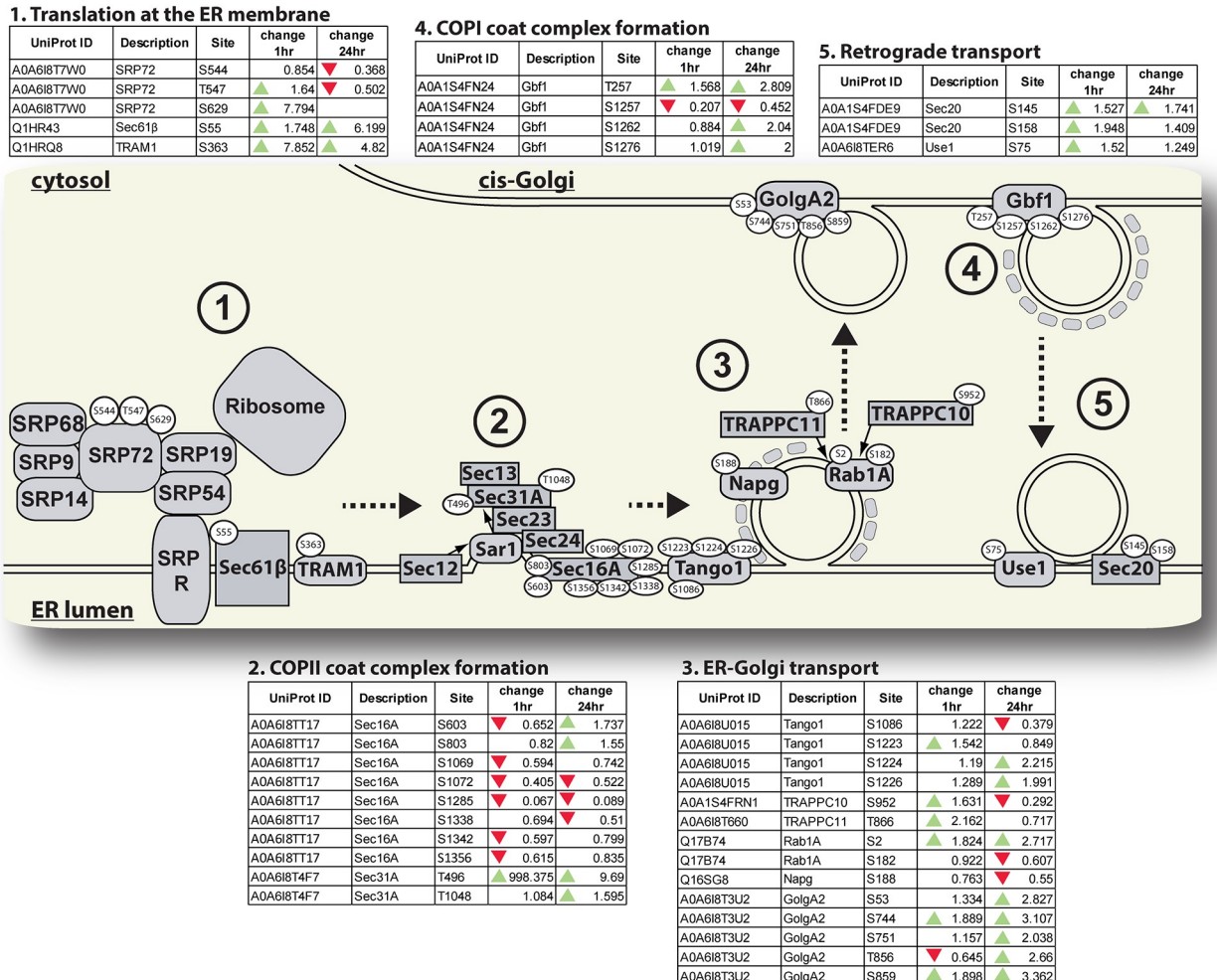

**2. COPII coat complex formation**

| UniProt ID | Description | Site | change 1hr | | change 24hr | |
|---|---|---|---|---|---|---|
| A0A6I8TT17 | Sec16A | S603 | 0.652 | ▼ | 1.737 | ▲ |
| A0A6I8TT17 | Sec16A | S803 | 0.82 | | 1.55 | ▲ |
| A0A6I8TT17 | Sec16A | S1069 | 0.594 | ▼ | 0.742 | |
| A0A6I8TT17 | Sec16A | S1072 | 0.405 | ▼ | 0.522 | ▼ |
| A0A6I8TT17 | Sec16A | S1285 | 0.067 | ▼ | 0.089 | ▼ |
| A0A6I8TT17 | Sec16A | S1338 | 0.694 | | 0.51 | ▼ |
| A0A6I8TT17 | Sec16A | S1342 | 0.597 | ▼ | 0.799 | |
| A0A6I8TT17 | Sec16A | S1356 | 0.615 | | 0.835 | |
| A0A6I8T4F7 | Sec31A | T496 | 998.375 | ▲ | 9.69 | ▲ |
| A0A6I8T4F7 | Sec31A | T1048 | 1.084 | | 1.595 | ▲ |

**3. ER-Golgi transport**

| UniProt ID | Description | Site | change 1hr | | change 24hr | |
|---|---|---|---|---|---|---|
| A0A6I8U015 | Tango1 | S1086 | 1.222 | | 0.379 | ▼ |
| A0A6I8U015 | Tango1 | S1223 | 1.542 | ▲ | 0.849 | |
| A0A6I8U015 | Tango1 | S1224 | 1.19 | | 2.215 | ▲ |
| A0A6I8U015 | Tango1 | S1226 | 1.289 | ▲ | 1.991 | ▲ |
| A0A1S4FRN1 | TRAPPC10 | S952 | 1.631 | ▲ | 0.292 | ▼ |
| A0A6I8T660 | TRAPPC11 | T866 | 2.162 | ▲ | 0.717 | |
| Q17B74 | Rab1A | S2 | 1.824 | ▲ | 2.717 | ▲ |
| Q17B74 | Rab1A | S182 | 0.922 | ▼ | 0.607 | ▼ |
| Q16SG8 | Napg | S188 | 0.763 | | 0.55 | ▼ |
| A0A6I8T3U2 | GolgA2 | S53 | 1.334 | | 2.827 | ▲ |
| A0A6I8T3U2 | GolgA2 | S744 | 1.889 | ▲ | 3.107 | ▲ |
| A0A6I8T3U2 | GolgA2 | S751 | 1.157 | | 2.038 | ▲ |
| A0A6I8T3U2 | GolgA2 | T856 | 0.645 | ▼ | 2.66 | |
| A0A6I8T3U2 | GolgA2 | S859 | 1.898 | ▲ | 3.362 | ▲ |

**Fig 3. Changes in phosphorylation of proteins associated with co-translation and ER-Golgi vesicle traffic.** *Ae. aegypti* proteins associated with co-translation and ER-Golgi transport conserved in other eukaryotic organisms are illustrated. Specific amino acid residues on proteins with phosphorylation changes of at least 1.5-fold between unfed and blood-fed mosquitoes are shown in white ovals. The numbers in the figure correspond to tables where proteins with related functions are grouped. Tables contain the Uniprot IDs for each protein as well as the fold-change in phosphorylation of each residue between unfed and either 1hr PBM or 24hr PBM mosquitoes. Green arrows represent an increase of at least 1.5-fold in phosphorylation at the specified residue between unfed and PBM mosquitoes, and red arrows represent a decrease of at least 1.5-fold in phosphorylation at the specified residue between unfed and PBM mosquitoes.

We also identified a large number of phosphoproteins involved in post-Golgi secretion and recycling processes. These include proteins involved in recruitment of clathrin and budding of clathrin-coated vesicles (Fig 4). We also identified changes in phosphorylation in proteins associated with endosomal sorting and recycling to the membrane, including Rab11b and its effector Rab11FIP5 (Fig 4). We identified changes in phosphorylation of proteins which regulate vesicle-membrane fusion including Synaptobrevin and Syntaxin B5 (Fig 4). Additionally, we found changes in phosphorylation of two kinesins and dynein associated proteins (Fig 4).

## TOR signaling cascade proteins

We observed changes in phosphorylation state of many proteins involved the nutrient-sensitive TOR signaling pathway (Fig 5). These include increased ratios of phosphorylated to unphosphorylated TSC1 and TSC2 at both PBM time points relative to unfed mosquitoes

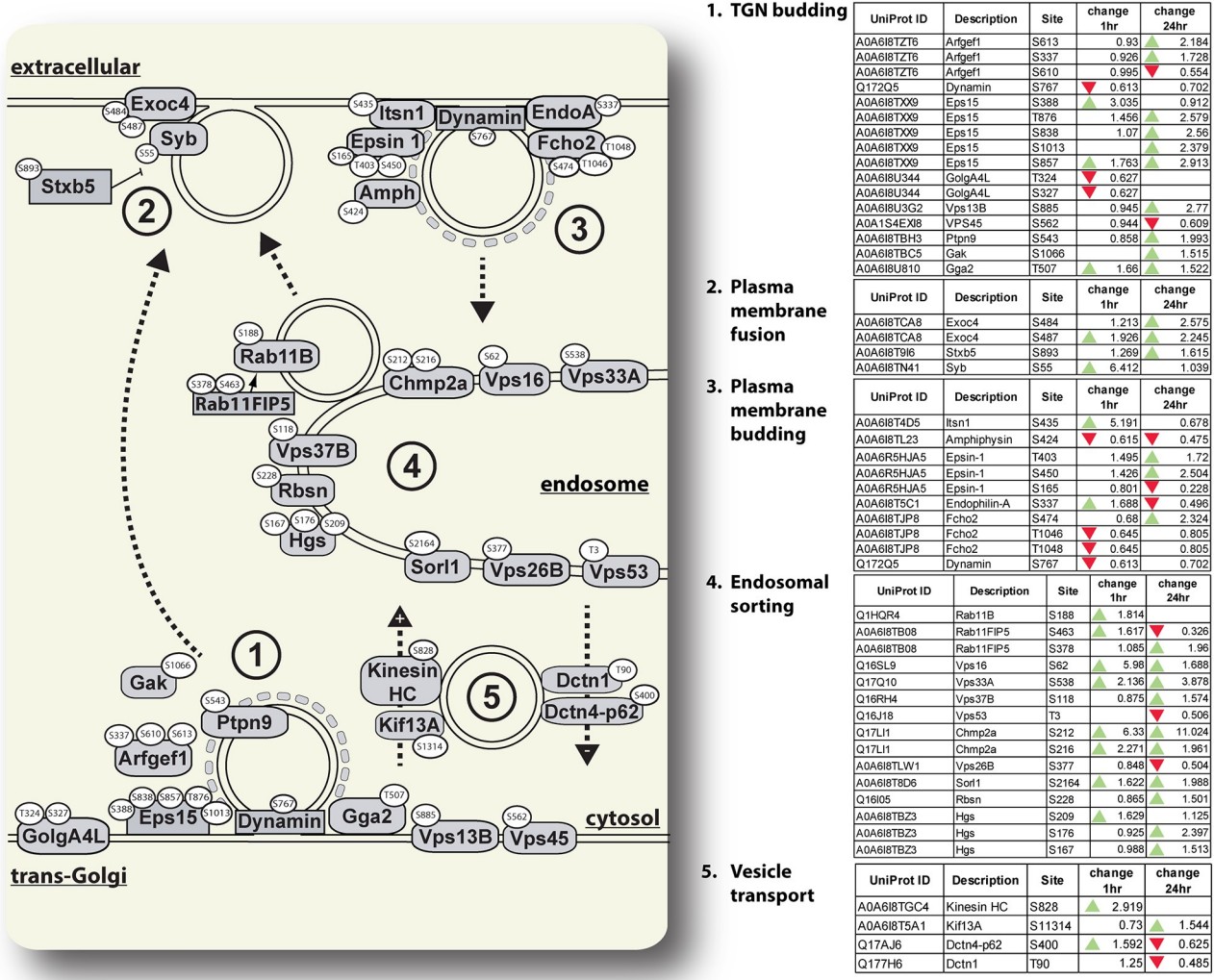

**Fig 4. Changes in phosphorylation of proteins associated with post-Golgi vesicle traffic.** *Ae. aegypti* proteins associated with traffic between the trans-Golgi, cell membrane, and endosome conserved in other eukaryotic organisms are illustrated. Specific amino acid residues on proteins with phosphorylation changes of at least 1.5-fold between unfed and blood-fed mosquitoes are shown in white ovals. The numbers in the figure correspond to tables where proteins with related functions are grouped. Tables contain the Uniprot IDs for each protein as well as the fold-change in phosphorylation of each residue between unfed and either 1hr PBM or 24hr PBM mosquitoes. Green arrows represent an increase of at least 1.5-fold in phosphorylation at the specified residue between unfed and PBM mosquitoes, and red arrows represent a decrease of at least 1.5-fold in phosphorylation at the specified residue between unfed and PBM mosquitoes.

(Fig 5a). We also observed increased phosphorylation of Sestrin and decreased phosphorylation of GATOR2 at both PBM time points relative to unfed mosquitoes (Fig 5a). Downstream of Tor kinase, we observed changes in phosphorylation of both S6 Kinase (S6K) and ribosomal protein S6 (S6) (Fig 5a).

Eukaryotic S6 proteins contain many phosphorylation sites [49, 50], so we performed a multiple sequence alignment of a set of S6 proteins representing a variety of eukaryote orders in COBALT [51] (Fig 5b) and identified a set of known mammalian phosphorylation sites which are conserved across many or all the species we included (Fig 5c). Of the four identified S6 serine residues with altered phosphorylation patterns PBM, three of these (S231, S235, and S237) are conserved with known phosphorylation sites in mammalian S6 (Fig 5c), giving us

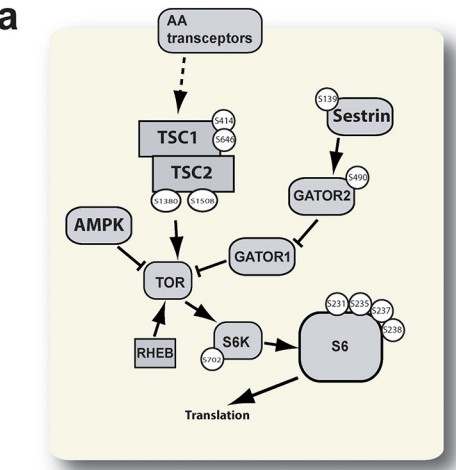

| UniProt ID | Description | Site | change 1 hr | Change 24 hrs |
|---|---|---|---|---|
| A0A6I8T6Q1 | TSC1 | S414 | 1.7 ▲ | 1.5 ▲ |
| A0A6I8T6Q1 | TSC1 | S646 | 1.1 | 1.5 ▲ |
| A0A6I8TP59 | TSC2 | S1380 | 18.2 ▲ | 9.3 ▲ |
| A0A6I8TA86 | TSC2 | S1508 | 7.0 ▲ | 8.8 ▲ |
| Q1HRW3 | Sestrin | S139 | 9.0 ▲ | 2.7 ▲ |
| Q175W5 | GATOR2 | S490 | 0.62 ▼ | 0.66 ▼ |
| A0A1S4F7R6 | S6K | S702 | 1 | 1.9 ▲ |
| Q5D0W4 | S6 | S231 | 17.4 ▲ | 70.6 ▲ |
| Q5D0W4 | S6 | S235 | 0.55 ▼ | 11.2 ▲ |
| Q5D0W4 | S6 | S237 | 75.0 ▲ | 7.7 ▲ |
| Q5D0W4 | S6 | S238 | 7.5 ▲ | 5.9 ▲ |

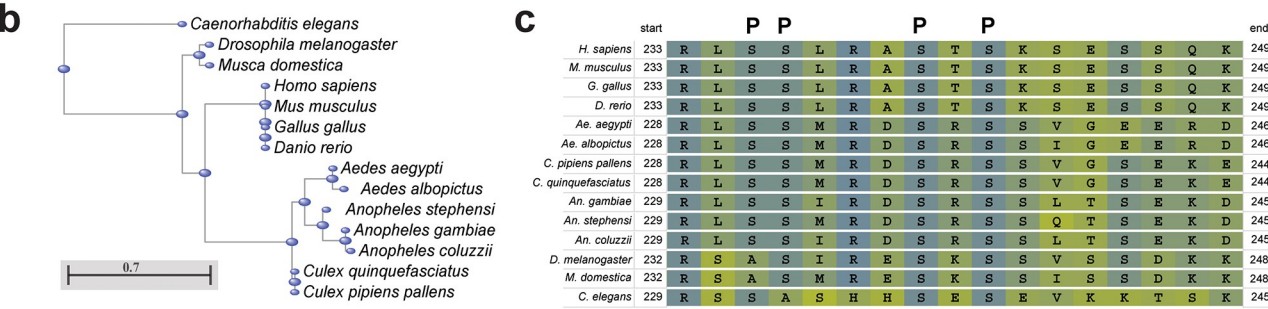

**Fig 5. Changes in TOR signaling, and conservation of rps6 phosphorylation sites. a.** many proteins involved in Tor signaling underwent changes in phosphorylation after blood feeding. The diagram shows important proteins in the *Ae. aegypti* Tor signaling pathway conserved in other eukaryotic organisms, with the specific residue(s) with changed phosphorylation state PBM. The table contains more information about each phosphorylated protein, including the Uniprot ID, and the change in phosphorylation of each residue at 1hr PBM relative to unfed, and at 24hr PBM relative to unfed. Green arrowheads represent at least 1.5-fold increased number of phosphorylated residues detected relative to the same position in unfed samples, and red arrowheads represent at least 1.5-fold decreased number of phosphorylated residues detected relative to the same position in unfed samples. **b.** Neighbor joining tree generated by the COBALT multiple sequence alignment tool (**Papadopoulos & Agarwala, 2007**), which illustrates the conservation of S6 in several orders of eukaryotes. **c.** Multiple sequence alignment of a portion of the S6 protein which is enriched in phosphorylation sites. This region contains four serine residues that are phosphorylated in mammalian S6 (marked by "P" above the alignment). All four of these residues are conserved in *Ae. aegypti*, and three of them (S231, S235, and S237) underwent changes in phosphorylation in MTs after blood feeding.

confidence that our phosphoproteomics dataset has provided an accurate picture of phosphorylation events in the MTs of *Ae. aegypti* before and after blood feeding.

In addition to proteins involved in mTOR signaling, we also identified several amino acid transporters with differential phosphorylation patterns before and after blood feeding (S1 File) including members of the excitatory amino acid transporter and proton-coupled amino acid transporter families.

## Cell adhesion proteins and paracellular transport

In addition to transcellular transport, water, ions and waste products may also enter the MT lumen via paracellular movement [52]. Therefore, we also looked for changes in phosphorylation of proteins associated with cell-cell junctions, and anchoring junctions to the cytoskeleton [12, 20]. We identified changes in phosphorylation of proteins associated with transmembrane linking of septate and subapical region junctions, as well as in proteins associated with anchoring linking proteins to the cytoskeleton (Fig 6). Interestingly, many of these changes represent decreases in phosphorylation relative to unfed mosquitoes. Of particular interest, we identified

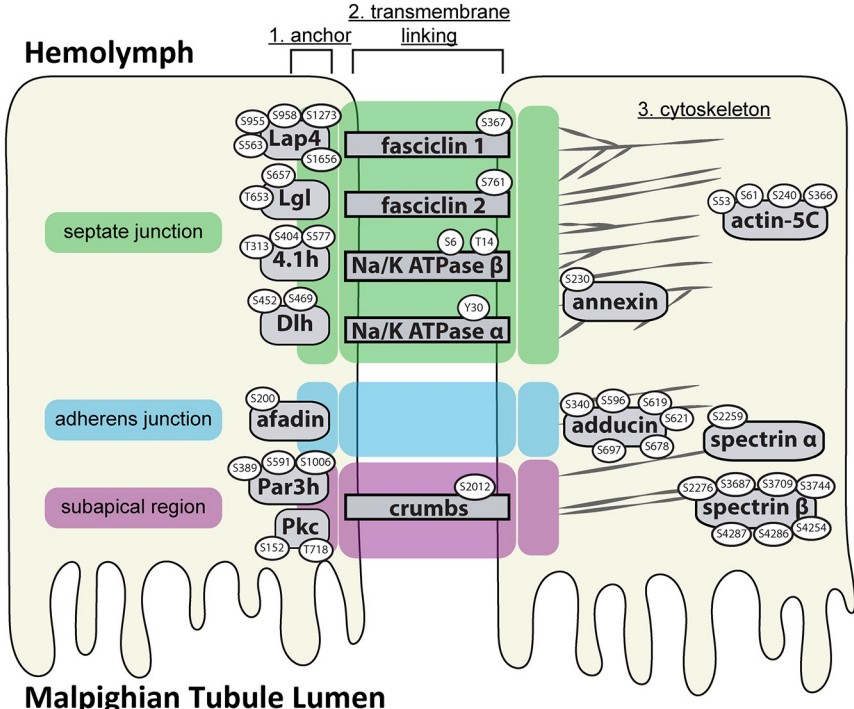

**1. intracellular anchor proteins**

| UniProt ID | Description | Site | Change 1 hr | Change 24 hr |
|---|---|---|---|---|
| A0A6I8U340 | Protein 4.1 homolog | T313 | ▼ 0.419 | ▼ 0.448 |
| A0A6I8U340 | Protein 4.1 homolog | S404 | 1.143 | ▲ 1.696 |
| A0A6I8U340 | Protein 4.1 homolog | S577 | 1.244 | ▲ 1.573 |
| A0A6I8TFN2 | afadin | S200 | 1.431 | ▼ 0.547 |
| A0A6I8TK32 | Disks large (Dlh) | S452 | 0.883 | ▼ 0.214 |
| A0A6I8TK32 | Disks large (Dlh) | S469 | ▼ 0.643 | 0.714 |
| A0A6I8TS30 | protein lap4 | S563 | 1.275 | ▲ 2.281 |
| A0A6I8TS30 | protein lap4 | S955 | ▲ 1.636 | ▲ 1.723 |
| A0A6I8TS30 | protein lap4 | S958 | ▲ 1.636 | ▲ 1.723 |
| A0A6I8TS30 | protein lap4 | S1273 | ▲ 2.7 | |
| A0A6I8TS30 | protein lap4 | S1656 | ▼ 0.651 | 0.862 |
| A0A1S4FP71 | lethal giant larva homolog (Lgl) | T653 | 1.077 | ▼ 0.406 |
| A0A1S4FP71 | lethal giant larva homolog (Lgl) | S657 | 0.933 | ▼ 0.523 |
| A0A6I8TEI2 | Par3 homolog | S389 | | ▼ 0.229 |
| A0A6I8TEI2 | Par3 homolog | S591 | | ▼ 0.308 |
| A0A6I8TEI2 | Par3 homolog | S1006 | 0.981 | ▼ 0.322 |
| A0A6I8T413 | Protein kinase C | S152 | 1.137 | ▼ 0.519 |
| A0A6I8T413 | Protein kinase C | T718 | 1.067 | ▼ 0.451 |

**2. transmembrane linking proteins**

| UniProt ID | Description | Site | Change 1 hr | Change 24 hr |
|---|---|---|---|---|
| A0A6I8U0Z3 | fasciclin 1 | S367 | ▼ 0.637 | ▼ 0.483 |
| Q16WM8 | fasciclin 2 | S761 | ▼ 0.621 | |
| J9EAN8 | Na/K ATPase β | S6 | ▼ 0.123 | ▼ 0.044 |
| Q16TS1 | Na/K ATPase β | T14 | ▲ 1.577 | ▲ 2.979 |
| A0A6I8TLQ1 | Na/K ATPase α | Y30 | 1.371 | ▼ 0.057 |
| A0A6I8TGR7 | crumbs | S2012 | 0.841 | ▲ 1.779 |

**3. cytoskeleton**

| UniProt ID | Description | Site | Change 1 hr | Change 24 hr |
|---|---|---|---|---|
| Q16QR7 | actin-5C | S53 | 1.147 | ▼ 0.561 |
| Q16QR7 | actin-5C | S61 | 0.823 | ▼ 0.656 |
| Q16QR7 | actin-5C | S240 | 0.693 | ▼ 0.456 |
| Q16QR7 | actin-5C | S366 | 0.764 | 0.48 |
| A0A1S4FSQ2 | adducin | S340 | ▼ 0.601 | 0.917 |
| M9NCK7 | adducin | S596 | 0.899 | ▲ 2.098 |
| M9NCK7 | adducin | S619 | ▼ 0.312 | 1.438 |
| M9NCK7 | adducin | S621 | 1.255 | ▲ 2.21 |
| M9NCK7 | adducin | S678 | ▼ 0.65 | 0.701 |
| M9NCK7 | adducin | S697 | | ▼ 0.135 |
| A0A6I8TMC7 | annexin | S230 | ▼ 0.582 | ▼ 0.565 |
| Q16EQ1 | spectrin α | S2259 | ▲ 2.573 | ▲ 1.593 |
| Q17SJ6 | spectrin β | S2276 | 0.962 | ▼ 0.551 |
| A0A6I8U5P4 | spectrin β | S3687 | 0.789 | ▼ 0.575 |
| A0A6I8U5P4 | spectrin β | S3709 | 1.188 | ▼ 0.415 |
| A0A6I8U5P4 | spectrin β | S3744 | 1.128 | ▼ 0.202 |
| A0A6I8U5P4 | spectrin β | S4254 | 1.447 | ▲ 1.566 |
| A0A6I8U5P4 | spectrin β | S4286 | ▼ 0.573 | 1.433 |
| A0A6I8U5P4 | spectrin β | S4287 | ▲ 1.741 | ▲ 1.766 |

**Fig 6. Changes in phosphorylation of cell-cell adhesion proteins.** *Ae. aegypti* proteins associated with cell junctions conserved in other eukaryotic organisms are illustrated (reviewed in **Beyenbach & Piermarini, 2011**). Specific amino acid residues on proteins with phosphorylation changes of at least 1.5-fold between unfed and blood-fed mosquitoes are shown in white ovals. Proteins in green highlighted regions are associated with septate junctions. Proteins in blue highlighted regions are associated with adherens junctions. Proteins in purple regions are associated with subapical region junctions. Numbers in the figure correspond to tables where proteins with related functions are grouped. Tables contain the Uniprot IDs for each protein as well as the fold-change in phosphorylation of each residue between unfed and either 1hr PBM or 24hr PBM mosquitoes. Green arrows represent an increase of at least 1.5-fold in phosphorylation at the specified residue between unfed and PBM mosquitoes, and red arrows represent a decrease of at least 1.5-fold in phosphorylation at the specified residue between unfed and PBM mosquitoes.

changes in phosphorylation in several proteins associated with spectrin and actin, which are involved in maintenance of cell membrane stability and cell shape and therefore may cause alteration of paracellular permeability, including Spectrin α and β chains, Actin-5C, Adducin, and Protein 4.1 Homolog [12, 20].

## Discussion

The Malpighian tubules of insects were named in honor of the 17[th] century Italian scientist Marcello Malpighi who discovered pulmonary capillaries and alveoli in vertebrates as well as spiracles in insects [53]. Studied for centuries, their development, anatomy, and function was thought to be well understood [54]. However, using new -omics approaches, researchers have recently identified a large number of genes/proteins expressed in the MTs that suggest novel functions in metabolism, detoxification, immunity, and other cellular and organismal processes [13]. Our present exploratory study adds new information by identifying thousands of phosphopeptides that are present in the Malpighian tubules of mosquitoes and by showing changes in their phosphorylation patterns after a blood meal. Phosphorylation of proteins is a posttranslational modification that has been shown to affect protein stability, protein/protein interactions, and protein function [55, 56]. Kinases and phosphatases regulate basically every aspect of cellular life [57]. Therefore, phosphoproteomics analysis of mosquito Malpighian tubules does not only provide an opportunity to develop hypothesis of how known processes are regulated within the cells of the tubules but also to identify new ones.

From a total of 1955 phosphoproteins with 4663 unique phosphorylation sites, the majority did not show any substantial changes in phosphorylation when comparing the unfed sample with the samples taken one and 24 hours after a blood meal. About 800 phosphoproteins showed changed phosphorylation patterns at the one hour time point. Interestingly, at the 24 hour time point the number had increased to approximately double this number. We hypothesize that this massive change in phosphorylation patterns reflect the activation and deactivation of cellular signaling, metabolic, and structural proteins as that are involved in the cellular processes shown in Figs 2–6.

Based on our findings the following picture emerges:

**V-ATPase subunits are dephosphorylated at 24 hours PBM**—According to the model of MT diuresis proposed by Beyenbach, the ion gradients required to power the osmotic flux of water into the MT lumen are powered by an electrochemical gradient created by vacuolar-type ATPase (V-ATPase) [7, 12, 13]. V-ATPases are evolutionary conserved multi subunit membrane proteins with two structural domains, the intracellular $V_1$ and the membrane standing $V_0$ domain [58]. While we did not observe any differences in phosphorylation of V-ATPase subunits between unfed and 1 hr PBM mosquitoes, we observed seven residues distributed across four $V_1$ subunits (B, E, H, F) with decreased phosphorylation at 24 hrs PBM relative to the unfed time point (Fig 2d). Interestingly, phosphorylation of V-type ATPase subunit C has previously been implicated in the assembly of $V_1$ and $V_0$ units in insects [59]. While we did identify phosphorylated residues in subunit C in our MT datasets (S1 File), none of these phosphorylation sites had changed phosphorylation patterns over the course of our experiment. Interestingly, a previous study on V-ATPase activity in *Ae. aegypti* MTs concluded that while Protein Kinase A activity was necessary for V-ATPase activation, direct phosphorylation of V-ATPase subunits may not be required for V-ATPase activation [47]. Therefore, V-ATPase activity in *Ae. aegypti* MTs may be regulated in a different fashion than the most well-known method of subunit C phosphorylation. We propose that the observed reductions in phosphorylation of V-ATPase subunits at 24 hrs PBM is responsible for the decrease in urine excretion within 24 hrs PBM that have been observed in *Ae. aegypti* and other mosquito species [8, 11, 41, 60]. It is also possible that more rapid or transient phosphorylation of subunit C happens in *Ae. aegypti* MTs prior to 1 hr PBM, and we did not observe these changes in the time points investigated in this study. Finally, it is possible that changes in phosphorylation of the subunits we identified do not play a direct role altering V-ATPase activity but may have other effects such as

mediating interactions between the subunits and other cytosolic or organellar proteins. Experiments to determine what effects of dephosphorylation of these subunits have on V-ATPase activity, and whether these dephosphorylation events produce the same effects as those seen after changes in subunit C phosphorylation [59] are necessary.

**Na+/H+ exchangers have altered phosphorylation patterns at early and later times PBM—** We observed changes in phosphorylation of two Na$^+$/H$^+$ exchangers (NHE3 and NHA2) by 1 hr PBM (Fig 2). NHA2 has been shown to localize to the apical membrane of SCs in *Anopheles gambiae* [61], while NHE3 localizes to the basolateral membrane of MT cells [62]. Interestingly, *An. gambiae* NHA2 gene expression was shown to decrease 3 hrs PBM [63], which was hypothesized to be due to NHA2 being used to preserve hemolymph sodium content, and therefore having reduced expression during natriuresis [61, 63]. We detected an increase in NHA2 phosphoprotein levels at 1 hr PBM, and a decrease in NHA2 phosphoprotein levels by 24hrs PBM (Fig 2). Our data provides support for this hypothesis that NHA2 activity plays an important role in sodium homeostasis by likely changing activity at different times during diuresis as Na$^+$ excretion rates change. NHE3 phospho-S770 increased 187-fold at 1 hr PBM, while phospho-S772 decreased 167-fold at this time point. We hypothesize that the shifting phosphorylation patterns of these two phosphorylation sites could be involved in the regulation of the activity of this exchanger and might represent an 'on-off' switch for this protein. NHE isoforms in in other eukaryotic species have been demonstrated to be activated by phosphorylation in response to growth factor signaling [64, 65]. Hormone signaling plays an important role in initiating diuresis after blood feeding [8, 17–19], so it is possible that DH31-stimulated intracellular signaling pathways in MT epithelia are responsible for the phosphorylation of NHE3 we observed in our samples. Previous work characterizing the localization and function of *Ae. aegypti* NHE3 identified a predicted set of phosphorylation sites, with many enriched in the intracellular C-terminus [62]. Our phosphoproteome analysis revealed a similar enrichment of phosphorylated amino acid residues in the C-terminus, but surprisingly, only one of the predicted phosphorylation sites from that study (S1017) changed phosphorylation in our study. We found only one other site from Pullikuth and colleagues' NHE3 characterization (S1099) in our dataset. The C-terminus of NHE3 appears to be a rich region of possible phosphorylation sites as demonstrated by predictions [62] and by our analysis. Future studies to validate the differential phosphorylation of S770 and S772, as well as the S1017 residue which was both predicted by Pullikuth et al. [62] and found in our dataset will be useful for understanding how the activity of this important transporter is regulated after blood feeding.

**Aquaporins are dephosphorylated at the 24 hr PBM time point—** We did not observe a corresponding change in AQP phosphorylation state at 1 hr PBM. We interpret this finding as support for the hypothesis that mosquito AQPs are not regulated by gating at the time points we studied as has been shown in the regulation of plant AQPs [29]. This would be consistent with the finding that human AQP2 C-terminal phosphorylation does not affect its water transport [66]. We acknowledge that these data do not preclude gating by phosphorylation having a more immediate effect on water transport earlier than 1 hr PBM.

**Phosphorylation of vesicle transport proteins changes dynamically PBM—** We observed changes in phosphorylation of many proteins associated with co-translation and transmembrane domain insertion at the ER, COPII vesicle formation, and ER-Golgi transport (Fig 3). Of particular interest, we observed an almost 1000-fold increase in phosphorylation of the T496 residue of the COPII coat protein, Sec31A at 1 hr PBM (Fig 3). Previous research has shown that phosphorylation of this protein plays an important role in regulating ER-Golgi

vesicle transport in yeast and human cells [67, 68], so our observation of extremely increased phosphorylation after blood feeding provides confidence in our conclusion that vesicle transport between organelles and to plasma membranes of MT cells increases PBM. In addition to regulation of proteins involved in new membrane/secretory protein synthesis, we observed changes in a variety of endosomal proteins involved in sorting recycled contents into other compartments. In particular, several proteins with altered phosphorylation are involved in sorting endosomal contents into lysosomes and multivesicular bodies (MVBs). The alterations in phosphorylation of proteins involved in endosomal sorting to lysosomes and MVBs (Fig 4) are interesting, as they do not directly influence solute or ion transport. However, there are several important roles that these proteins may play during diuresis. First, they may be important for sorting damaged membrane proteins to prevent them from being recycled to the cell membrane where they would affect the efficiency of water and solute secretion. Second, they may play important roles in sorting waste products taken in from the basolateral membrane to lysosomes or MVBs to be detoxified and cleared without causing damage to the MT cells. Research on the ground beetle, *Calathus fuscipes* has shown that MVB accumulation and exocytosis occurs in MTs of this insect [69], which provides support for our second proposed mechanism. Interestingly, a previous study in a related mosquito species, *Ae. taeniorhynchus* determined that by 24 hrs PBM, the microvilli brush border structure had reduced when compared to unfed mosquitoes [70]. We propose that this may be due to changes in membrane allocation, with brush border membranes having large amounts of microvilli to provide surface area for fluid and solute exchange, while later after blood feeding, more membrane structures may be dedicated to endosomes, lysosomes and MVBs for detoxification and secretion of waste products.

**Proteins of the TOR signaling cascade are phosphorylated through 24 hr PBM**—The TOR signaling pathway is part of a cellular nutrient sensing system and involved in regulating protein synthesis [2, 71–73]. Several upstream regulators of the TOR kinase, as well as the TOR kinase substrate, S6 kinase, and its substrate, ribosomal protein S6 are enriched with phosphorylation sites that are hyperphosphorylated PBM (Fig 5a). Ribosomal protein S6 phosphorylation is thought to play an important role in new protein synthesis [2, 72–74], and we observed changes in phosphorylation of four serine residues, three of which are conserved across all of the insects and vertebrates we analyzed (Fig 5c). The hyperphosphorylation of proteins of the TOR signaling cascade remained through 24 hrs PBM (Fig 5a).

We hypothesize that TOR signaling likely is activated to initiate gene expression to facilitate the expression of genes to support the need of cells for proteins to continue diuretic and detoxification activities. Previous research in *An. gambiae* MTs has shown increased expression of proteins with putative involvement in amino acid transport after blood feeding [63]. Additionally, we observed changes phosphorylation of proton-coupled amino acid transport proteins (S1 File), which have previously been shown to regulate mTOR signaling [75]. These results suggest that the cells of mosquito MTs have a nutrient sensor that involves the TOR signaling cascade and that this cascade is activated in the MTs after a blood meal. To the best of our knowledge, this study is the first to report on the mTOR signal pathway in mosquito Malpighian tubules.

**Changes in phosphorylation of cell junction proteins could affect paracellular water flow PBM**—We also observed alterations in phosphorylation of several proteins associated with cell-cell junctions at 1 hr PBM (Fig 6), indicating that the structure of these junctions may change after blood-feeding. While some phosphorylation changes occurred at 1 hr PBM, overall, more dephosphorylation of cell junction proteins occurred at the 24 hr PBM time

point. Previous research has demonstrated that several phosphoproteins identified in our study dynamically re-localize out of the cytoplasm of MT cells after treatment with the diuretic hormone aedeskinin-III [76]. Additionally, there is evidence that actin and adducin phosphorylation changes in response to natriuretic hormone signaling [76–78]. These studies suggest that phosphorylation of these proteins allow for remodeling of the actin-spectrin cytoskeleton, which may alter PC apical brush border membrane structure or paracellular transport [76, 77]. Our findings mesh well with these previous studies, as we observed different phosphorylation patterns in these proteins after blood feeding. In particular, actin was demonstrated to re-locate to the apical brush border of PCs in response to cyclic AMP treatment relative to non-treated PCs [78]. In this study, we observed dephosporylation of Actin-5C by 24 hr PBM (Fig 6), indicating that actin filaments may be depolymerizing at this time point. This is also supported by our findings of adducin phosphorylation, as phosphorylation of adducin leads to disassociation of actin and spectrin, and we observed reduced adducin phosphorylation at 1 hr PBM and enrichment of adducin phosphorylation at 24 hrs PBM (Fig 6). This indicates that even though much of the water from the blood meal has been excreted, the organization of cell-cell junctions remains dynamic through the first 24 hr PBM.

**Limitations of this study**—The results reported in this paper are based on single samples containing pools of MTs from 200 mosquitoes. We acknowledge that a single sample for each time point elevates the risk of outliers affecting our data and does not allow for statistical analysis of our dataset. However, we believe our data represent a robust phosphoprotein profile of MTs at each time point, and that pools of MTs from 200 individuals will likely minimize the chance of one or a few outlier MTs having an outsized effect on the overall data from any sample. Additionally, we reported on pathways and processes with many associated proteins that underwent changes in phosphorylation post-blood meal, and therefore the likelihood that we identified changes in phosphoproteins with real biological significance is high, despite the lack of sample size. We also acknowledge that without a proteome sample for each time point, we cannot state with certainty that alterations in phosphorylation are due solely to kinase/phosphatase activity on existing proteins, or if some of the observed changes are due to changes in synthesis/degradation of particular proteins. Future studies to validate how much of our observed changes are due to altered phosphorylation of existing proteins, and how much are due to new protein synthesis will prove useful to validate these data. These validation experiments will be performed in future studies based on these data. We chose not to sample mosquitoes from any time point prior to 1 hr PBM to ensure that we could standardize our samples. As diuresis begins within minutes of the initiation of a blood meal and urine production can last up to three hours PBM, we believe that selecting 1 hr PBM as our "early" time point gives an accurate profile of early protein regulation, but we acknowledge that phosphorylation events likely happen rapidly after initiation of blood feeding. This problem could be solved with the use of a blood feeding system allowing for blood feeding and sampling of individual mosquitoes. Kumar et al. recently described a 3D-printed chip system that allows for feeding and monitoring of individual mosquitoes [79], which would prove useful for future experiments to assay for rapid changes in protein phosphorylation within the first hour of blood feeding.

## Conclusion

Taken together, our phosphoproteome analysis provides evidence that supports the model of MT activity proposed by Beyenbach and colleagues. We suggest that AQP activity in insect

MT is not regulated by gating but rather shuttling and changes in gene expression. The phosphoproteome data generated in this exploratory study both represent a preliminary picture of the processes that occur in female *Ae. aegypti* after a blood meal. They provide a first baseline for the development of hypotheses and future studies that will involve pharmaceutical and molecular manipulations of individual proteins.

## Supporting information

**S1 File. Table containing all phosphoproteomics data.** Phosphoproteomics dataset containing all phosphoproteins detected (Quant tab), mass spectrometry data for all phosphopeptides detected (pep_quant tab), and phosphoproteins with differential phosphorylation patterns (phosphorylation change of 1.5-fold increase or decrease) between unfed and post-blood meal mosquitoes (remaining tabs).
(XLSX)

**S2 File. Table containing phosphoproteins referenced in figures and manuscript text.** Descriptions of protein function were compiled from UniProt [80] and/or the GeneCards Human Gene Database (https://www.genecards.org) [81].
(XLSX)

**S3 File. Table containing phosphoproteins with greater than 25-fold increase in phosphorylation between unfed and at least one PBM sample.**
(XLSX)

## Acknowledgments

The following reagent was provided by the NIH/NIAID Filariasis Research Reagent Resource Center for distribution through BEI Resources, NIAID, NIH: Aedes aegypti, Strain Black Eye Liverpool, Eggs, NR-48921. The authors wish to thank Soumi Mitra, Marialusa Tiexiera and Conrad Brooks for assisting with Malpighian tubule dissections, and Hailey Luker and Carolyn Armendariz for assistance with mosquito rearing.

## Author Contributions

**Conceptualization:** Immo A. Hansen.

**Data curation:** Yashoda Kandel, Matthew Pinch, Nathan Martinez, Immo A. Hansen.

**Formal analysis:** Yashoda Kandel, Matthew Pinch, Immo A. Hansen.

**Funding acquisition:** Immo A. Hansen.

**Investigation:** Yashoda Kandel.

**Writing – original draft:** Yashoda Kandel, Matthew Pinch, Mahesh Lamsal, Immo A. Hansen.

**Writing – review & editing:** Yashoda Kandel, Matthew Pinch, Immo A. Hansen.

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
