## [Decision Letter · Decision Letter 0]

9 May 2022

PONE-D-22-09152Phosphoproteomics profiling of Aedes aegypti Malpighian tubules during blood meal processing reveals dramatic transition in functionPLOS ONE

Dear Dr. Hansen,

Thank you for submitting your manuscript to PLOS ONE. After careful consideration, we feel that it has merit but does not fully meet PLOS ONE’s publication criteria as it currently stands. Therefore, we invite you to submit a revised version of the manuscript that addresses the points raised during the review process.

Both reviewers acknowledged that your article has a substantive contribution and provides an innovative approach that can help and influence further work on this subject. They both made a detailed analysis of the manuscript and data. Rev #1 made an extensive list of suggestions that may help to improve the manuscript. However, some relevant issues that need to be responded were raised  by Rev #2. Specifically, try to address (1) Rev #2 demand on more detail on how differential expression was calculated, (2) rev #2 comments on data displayed in figures 3-6; (#) check if his question if “ massive values may reflect ‘divide by near zero’ artefacts” applies for some  results. Here I think he is particularly concerned about numbers as high as 70-fold variations in phosphorylation and (3) comment on the replication issues that were raised.

We look forward to receiving your revised manuscript.

Kind regards,

Pedro L. Oliveira

Academic Editor

PLOS ONE

Journal Requirements:

Reviewers' comments:

Reviewer's Responses to Questions

**Comments to the Author**

1. Is the manuscript technically sound, and do the data support the conclusions?

Reviewer #1: Yes

Reviewer #2: Partly

2. Has the statistical analysis been performed appropriately and rigorously? 

Reviewer #1: Yes

Reviewer #2: I Don't Know

3. Have the authors made all data underlying the findings in their manuscript fully available?

Reviewer #1: Yes

Reviewer #2: Yes

4. Is the manuscript presented in an intelligible fashion and written in standard English?

Reviewer #1: Yes

Reviewer #2: Yes

5. Review Comments to the Author

Reviewer #1: Manuscript PONE-D-22-09152 by Kandel et al. is a highly novel study that investigates changes in the phosphoproteome of Malpighian tubules of Aedes aegypti after blood feeding. Malpighian tubules play particularly important roles in mediating a dramatic diuresis within the 1st hour after blood feeding and previous transcriptomic studies suggest they undergo a dramatic functional transition within 24 hours after blood feeding. However, insights into the roles of protein phosphorylation in either of these processes in mosquito Malpighian tubules is limited. Moreover, studies on protein phosphorylation in Malpighian tubules of any insect are very limited. Thus, the present study is highly innovative and provides new insights into the regulation of Malpighian tubule physiology. Also, the work is conducted on an important human disease vector, which adds potential applied significance to their findings. The only areas for improvement (mostly minor or moderate concerns) found by this reviewer are to improve the clarity of presenting some of the results and protein groups of interest (see below for details) and to add more depth to the interpretations of their findings in the physiological context of Malpighian tubules (see below for details). Below are specific comments aimed at helping the authors improve their manuscript:

p.4—The neurohormone mentioned in the 2nd paragraph is referred to now as the ‘calcitonin-like diuretic hormone or ‘diuretic hormone 31 (DH31) ’. The authors are encouraged to include the following citations of more recent publications that have further characterized this peptide and its receptor:

G.M. Coast, C.S. Garside, S.G. Webster, K.M. Schegg, D.A. Schooley, Mosquito natriuretic peptide identified as a calcitonin-like diuretic hormone in Anopheles gambiae (Giles), J Exp Biol, 208 (2005) 3281-3291.

H. Kwon, H.L. Lu, M.T. Longnecker, P.V. Pietrantonio, Role in Diuresis of a Calcitonin Receptor (GPRCAL1) Expressed in a Distal-Proximal Gradient in Renal Organs of the Mosquito Aedes aegypti (L.), PLoS One, 7 (2012) e50374.

K.A. Halberg, S. Terhzaz, P. Cabrero, S.A. Davies, J.A. Dow, Tracing the evolutionary origins of insect renal function, Nat Commun, 6 (2015) 6800.

p.4-5—The authors are encouraged to include mention of more recent publications on AQPs that have characterized immunolabeling in larval Aedes MTs, which may be useful in the introduction and discussion:

L. Misyura, E. Grieco Guardian, A.C. Durant, A. Donini, A comparison of aquaporin expression in mosquito larvae (Aedes aegypti) that develop in hypo-osmotic freshwater and iso-osmotic brackish water, PLOS ONE, 15 (2020) e0234892.

L. Misyura, G.Y. Yerushalmi, A. Donini, A mosquito entomoglyceroporin, Aedes aegypti AQP5, participates in water transport across the Malpighian tubules of larvae, Journal of Experimental Biology, 220 (2017) 3536-3544.

p.7—A rationale for the time points selected for the tissue samples (1h&24 h PBM) would be helpful in the Introduction. One is provided in the Discussion on p.18, but I would suggest moving this to the Intro or Methods so the reader better understands why those time points were selected.

p.12—It is not clear to this reviewer whether ‘differentially expressed proteins’ refers to a change in phosphorylation state, change in protein abundance, or a combination of both? It would be useful to better define this term to help the reader interpret the results.

p.12—I would suggest using ‘transport’ instead of ‘transportation’ in the text and all figures.

p.12—I would suggest using ‘powered’ instead of ‘mediated’ as the latter suggests the V-ATPase directly transports Na and K, which may confuse some readers.

p.12—Remove ‘channels’ after ‘antiporter’. Antiporters are a type of membrane transporter and not a channel.

p.13—‘sodium/hydrogen exchanger’ is too vague. The authors are showing changes to two specific NHE/NHA proteins in Fig. 2. One of them (A0A6I8TKG3) corresponds to an ortholog of Drosophila NHA1 (AAEL011109) and the other (A0A6I8T520) corresponds to an ortholog of Drosophila NHE2 (AAEL001503) also known as Aedes NHE3 (using nomenclature of Gill). Distinguishing these two proteins is important as they are expected to be expressed on different cell types and membranes in Malpighian tubules. See the following references for more details:

A.K. Pullikuth, K. Aimanova, W. Kang'ethe, H.R. Sanders, S.S. Gill, Molecular characterization of sodium/proton exchanger 3 (NHE3) from the yellow fever vector, Aedes aegypti, J Exp Biol, 209 (2006) 3529-3544.

M.A. Xiang, P.J. Linser, D.A. Price, W.R. Harvey, Localization of two Na+- or K+-H+ antiporters, AgNHA1 and AgNHA2, in Anopheles gambiae larval Malpighian tubules and the functional expression of AgNHA2 in yeast, J Insect Physiol, 58 (2012) 570-579.

p.13—Figure 2 legend—Beyenbach is misspelled.

p.14-15—The results sections on endomembrane and TOR contain several abbreviations and proteins that the reviewer was unfamiliar with (e.g., TSC1; sestrin; GATOR2). It may be useful to include a supplemental table that spells out each abbreviation used in these sections and provides a brief description of each protein’s known function or role in another system.

p.16-17—The authors indicate that changes in spectrin and actin were ‘of particular interest’, but it is unclear why they are of particular interest. More context or background may help.

p.18—As noted above, the paragraph explaining the rationale for the 3 time points may be more helpful in the Intro or Methods.

Discussion in general: More in depth interpretation of the significance of the phosphorylation on the biochemical activity of the specific protein groups would be welcomed for interpreting potential physiological consequences in the context of Malpighian tubule function. Most protein groups mentioned are discussed in general terms and repeat the presentation of results, but do not provide much in terms of physiological interpretation, especially in the context of how the tubules are thought to be functioning at the different time points measured. More speculation and generation of specific hypotheses is encouraged.

For example, on p.19, V-ATPase--- What is the significance of the dephosphorylation on V-ATPase activity? Others have shown that PKA enhances V-ATPase activity in Aedes MTs (e.g., F. Tiburcy, K.W. Beyenbach, H. Wieczorek, Protein kinase A-dependent and -independent activation of the V-ATPase in Malpighian tubules of Aedes aegypti, J Exp Biol, 216 (2013) 881-891.). Could dephosphorylation potentially explain in part the previously documented decrease in diuresis capacity observed by Esquivel et al (2016 PeerJ) 24 h PBM in Aedes albopictus?

Also on p.19, for NHE-- What effects does phosphorylation have on other NHEs (activation, deactivation)? What insights does this potentially provide into exchange activity across basolateral membrane for NHE3 and apical membrane for NHA2, and how ions are transporter? Also, confirm whether the NHA is NHA1 or 2. In the original paper that cloned Aedes NHE3, Pullikuth et al (2006) identified predicted phosphorylation sites. How do the observed phosphorylation sites in the present study compare with the predictions?

p. 19, aquaporins—The authors should clarify that the findings are consistent with the hypothesis at the time points measured. It is still possible that gating via phosphorylation could be involved at early time points not examined in this study (e.g., < 1h PBM).

p.20, vesicle transport proteins—The authors should clarify on what is meant by ‘membrane secretion’. This is a vague term. Are the authors referring to apical, basolateral, intracellular, or intercellular membranes? Secretion of ions, water, or wastes? How would the authors predict the changes to endosomal content sorting into lysosomes and MVBs affect tubule function? It would be interesting for the authors to discuss their findings in the context of a previous study by Bradley et al. that found ultrastructural changes in apical brush border of Aedes MTs 24 PBM:

T.J. Bradley, D.M.J. Sauerman, J.K. Nayar, Early cellular responses in the Malpighian tubules of the mosquito Aedes taeniorhynchus to infection with Dirofilaria immitis Nematoda, Journal of Parasitology, 70 (1984) 82-88.

p.20, TOR signaling—The idea of a nutrient sensor is very interesting, but how does it potentially relate to the context of MT function? More details explaining how TOR signaling would impact known physiological functions would be helpful. Is this the first study to identity a potential TOR signalling pathway in insect Malpighian tubules? More explanation of the significance of this finding is welcomed to highlight a potentially high impact finding.

p.20-21, cell-cell junctions—The authors should include a discussion of findings from earlier studies that provided proteomic insights into how intercellular junctions, the cytoskeleton, and apical membrane are modified by diuretic hormones involved with the post-blood meal diuresis (e.g., aedeskinin). One study in particular (Miyauchi et al. 2013) provided a detailed examination of the impact of phosphorylation of adducin that would be interesting to compare with the present findings. Another study by Karas et al. should be of interest to the authors given their interest in the actin-spectrin cytoskeleton. The authors are encouraged to cite these studies and incorporate them into the discussion where relevant.

J.T. Miyauchi, P.M. Piermarini, J.D. Yang, D.M. Gilligan, K.W. Beyenbach, Roles of PKC and phospho-adducin in transepithelial fluid secretion by Malpighian tubules of the yellow fever mosquito, Tissue Barriers, 1 (2013) 1-14.

K.W. Beyenbach, S. Baumgart, K. Lau, P.M. Piermarini, S. Zhang, Signaling to the apical membrane and to the paracellular pathway: changes in the cytosolic proteome of Aedes Malpighian tubules, J Exp Biol, 212 (2009) 329-340.

K. Karas, P. Brauer, D. Petzel, Actin redistribution in mosquito Malpighian tubules after a blood meal and cyclic AMP stimulation, J Insect Physiol, 51 (2005) 1041-1054.

p.21, conclusion—Although the present study provides a major advance in regards to the phosphoproteomics of MTs, the authors should include some caveats regarding the limitations of their study and emphasize that their findings are only relevant to the time points examined. Thus, they cannot rule out effects at earlier time points (e.g., 5 min PBM). Given that diuresis begins as soon as blood ingestion begins and post-translational changes can occur rapidly (within minutes) the authors may want to suggest future studies examine earlier time points PBM to provide further insights into relatively immediate changes in the MT phosphoproteome that may regulate the switch-like stimulation of diuresis after blood feeding (e.g., Beyenbach et al. 2009).

Figure 1-- The illustration of the mosquito is not described in the legend. Its context is unclear based on the legend and the text in which it is cited. Eliminate the mosquito or add a description. The cartoon of the different regulatory mechanisms is very clear and effective.

Figure 2b-- Only Na is shown to be antiported across the apical membrane in the figure. K+ should also be included to reflect what is stated in the text. Also, there is no evidence for apical K+ channels in principal cells of mosquito tubules. This should be removed from the model. There is evidence for basolateral K+ channels in principal and stellate cells (see reference below):

P.M. Piermarini, S.M. Dunemann, M.F. Rouhier, T.L. Calkins, R. Raphemot, J.S. Denton, R.M. Hine, K.W. Beyenbach, Localization and role of inward rectifier K+ channels in Malpighian tubules of the yellow fever mosquito Aedes aegypti, Insect Biochem Mol Biol, 67 (2015) 59-73.

Table S1—It would be convenient to the reader if the gene names (column E) could all be standardized to the vectorbase ID#s (e.g., AAEL) for facilitating follow ups on orthologs and available data in vectorbase.

Reviewer #2: This MS presents an interesting and thought-provoking dataset, and will be a useful contribution to the field.

Major points:

More detail is needed on how differential expression is scored. If, say, a protein doubles its expression level between t=1 and t=1h, but its percentage phosphorylation remains the same, would it show as the same, or 100% upregulated, in these data? Change in expression level is also a perfectly good way of regulating activity! The broader issue of normalization of data also needs to be addressed explicitly.

Like most ‘big data’ papers, the results are over-stated. I’m particularly exercised by figs 3-6. Where did these graphics come from? Are they BioRenders of human pathways? Are any of these proteins /pathways experimentally validated in Aedes? The legends need to be far more specific, along the lines “Human vesicle trafficking pathway (BioRender), together with Aedes homologs with significant phosphorylation differences. Proteins with upregulated phosphorylation at both time points are coloured green, down regulated red, and proteins with mixed phosphorylation are coloured red/green or as appropriate. Proteins not detected, or without significant changes, or with no Aedes homologs, are left unshaded.” Appropriately colour coding the figure would then be much more useful to the reader.

Similarly, the authors should go through the MS and check that they qualify their claims suitably.

Some of the changes are truly remarkable. The authors may want to bring these out into a separate table; but also carefully scrutinise the raw data to ensure that the proteins are truly expressed at all timepoints – otherwise, these massive values may reflect ‘divide by near zero’ artefacts.

Some of the more original points need independent validation; the data presented are based on a small number of runs from a commercial provider; did the authors try to independently validate any of these phosphorylation changes themselves?

Minor points:

Cross-referencing against microarray data. This could be taken a little further, and would be interesting. Although the authors cite Esquivel et al., they may also want to have a look at Overend et al (https://doi.org/10.1016/j.ibmb.2015.05.007) – although this paper was in Anopheles, the underlying microarray (Affymetrix) was much more comprehensive. And very recently, there’s another Aedes paper – Hixson et al (https://doi.org/10.7554/eLife.76132), that may be of use.

V-ATPases: The authors suggest that multiple subunits change their phosphorylation status. This is a big deal, because despite several decades of looking, the field is really satisfied with the data only for subunit C – which these authors find to be unchanged. Rather than focus on Aedes, they should back up their claims of phosphorylation of key targets with evidence to show that this is known to happen elsewhere. The V-ATPase, incidentally, is one example where direct independent verification would be welcome.

Statistics - I didn't notice it, but was Bonferroni or other correction made for multiple data?

6. PLOS authors have the option to publish the peer review history of their article (what does this mean?). If published, this will include your full peer review and any attached files.

Reviewer #1: No

Reviewer #2: No

---

## [Author Response · Author response to Decision Letter 0]

27 May 2022

Department of Biology, MSC 3AF 

New Mexico State University

PO Box 30001

Las Cruces, NM 88003-8001

Telephone: (575) 646-3611

FAX: (575) 646-5665

26th May, 2022

Dr. Pedro L. Oliveira

Academic Editor, PLoS ONE

Re: "Phosphoproteomics profiling of Aedes aegypti Malpighian tubules during blood meal processing reveals dramatic transition in function” (Manuscript ID: PONE-D-22-09152)

Dear Dr. Oliveira,

Per your instructions, we have addressed each of the comments and suggestions provided by the Reviewers on our manuscript (ID: PONE-D-22-09152) entitled “Phosphoproteomics profiling of Aedes aegypti Malpighian tubules during blood meal processing reveals dramatic transition in function." We have thoroughly updated the manuscript in the process of addressing the comments of the Reviewers. Their thorough evaluation of our manuscript is greatly appreciated, and we believe that our comprehensive changes made in the course of addressing the Reviewers’ comments has increased the quality of our manuscript. We hope you will find it suitable for publication in your journal at this time.

Sincerely,

Immo A. Hansen, Ph.D.

Associate Professor

Department of Biology

Institute of Applied Biosciences

MSC 3AF

New Mexico State University

Las Cruces, NM 88003

Phone: (575) 646-7719

Email: immoh@nmsu.edu

 

Response to the Editor:

We are grateful for the Reviewers’ comments and believe that our manuscript will now satisfy the Reviewer’s concerns.

We wish to notify you of minor changes made in the course of addressing the Reviewers’ comments:

1. We have added several phosphorylation sites to the ‘Sodium and Hydrogen transport’ table in Figure 2 that were missing from the figure previously and have updated all table names in this figure from ‘transportation’ to ‘transport’ in accordance with Reviewer 1’s suggestion.

2. The Reference section has been updated to include references added as we updated the manuscript.

3. We have made several minor changes to the Methods, Results and Discussion sections to add clarity in addition to changes made in response to the Reviewers. None of these changes alters the overall conclusions of the study.

4. We have updated the text to correct typos.

 

Detailed response to Reviewers’ comments:

Reviewer #1: Manuscript PONE-D-22-09152 by Kandel et al. is a highly novel study that investigates changes in the phosphoproteome of Malpighian tubules of Aedes aegypti after blood feeding. Malpighian tubules play particularly important roles in mediating a dramatic diuresis within the 1st hour after blood feeding and previous transcriptomic studies suggest they undergo a dramatic functional transition within 24 hours after blood feeding. However, insights into the roles of protein phosphorylation in either of these processes in mosquito Malpighian tubules is limited. Moreover, studies on protein phosphorylation in Malpighian tubules of any insect are very limited. Thus, the present study is highly innovative and provides new insights into the regulation of Malpighian tubule physiology. Also, the work is conducted on an important human disease vector, which adds potential applied significance to their findings. The only areas for improvement (mostly minor or moderate concerns) found by this reviewer are to improve the clarity of presenting some of the results and protein groups of interest (see below for details) and to add more depth to the interpretations of their findings in the physiological context of Malpighian tubules (see below for details). Below are specific comments aimed at helping the authors improve their manuscript:

We thank the Reviewer for their favorable assessment of our manuscript. We have addressed the Reviewer’s comments in detail below.

p.4—The neurohormone mentioned in the 2nd paragraph is referred to now as the ‘calcitonin-like diuretic hormone or ‘diuretic hormone 31 (DH31) ’. The authors are encouraged to include the following citations of more recent publications that have further characterized this peptide and its receptor:

G.M. Coast, C.S. Garside, S.G. Webster, K.M. Schegg, D.A. Schooley, Mosquito natriuretic peptide identified as a calcitonin-like diuretic hormone in Anopheles gambiae (Giles), J Exp Biol, 208 (2005) 3281-3291.

H. Kwon, H.L. Lu, M.T. Longnecker, P.V. Pietrantonio, Role in Diuresis of a Calcitonin Receptor (GPRCAL1) Expressed in a Distal-Proximal Gradient in Renal Organs of the Mosquito Aedes aegypti (L.), PLoS One, 7 (2012) e50374.

K.A. Halberg, S. Terhzaz, P. Cabrero, S.A. Davies, J.A. Dow, Tracing the evolutionary origins of insect renal function, Nat Commun, 6 (2015) 6800.

We have updated the name of the neurohormone in the manuscript, and have included the recommended citations.

p.4-5—The authors are encouraged to include mention of more recent publications on AQPs that have characterized immunolabeling in larval Aedes MTs, which may be useful in the introduction and discussion:

L. Misyura, E. Grieco Guardian, A.C. Durant, A. Donini, A comparison of aquaporin expression in mosquito larvae (Aedes aegypti) that develop in hypo-osmotic freshwater and iso-osmotic brackish water, PLOS ONE, 15 (2020) e0234892.

L. Misyura, G.Y. Yerushalmi, A. Donini, A mosquito entomoglyceroporin, Aedes aegypti AQP5, participates in water transport across the Malpighian tubules of larvae, Journal of Experimental Biology, 220 (2017) 3536-3544.

We have included the recommended citations.

p.7—A rationale for the time points selected for the tissue samples (1h&24 h PBM) would be helpful in the Introduction. One is provided in the Discussion on p.18, but I would suggest moving this to the Intro or Methods so the reader better understands why those time points were selected.

We have moved the paragraph mentioned by the Reviewer from the Discussion to the Introduction. The text now reads:

The goal of this study was to improve our understanding of the mechanisms that regulate ions and water balance in mosquitoes after a blood meal. We investigated changes in MT protein phosphorylation before and after blood feeding. The three time points we chose for our phosphoproteomics analysis cover the MTs in three very different states. The initial time point included unfed mosquito females that were approximately 72 hours after hatching and in a reproductive state-of-arrest (40). Little diuresis is happening during this stage (41). The second time point, one hour after a blood meal, covers the MT at their peak of activity during the rapid dehydration of the blood meal (6). At the third time point, the initial dehydration of the blood meal is completed but digestion of the blood proteins, hemoglobin and serum proteins, is in full swing producing toxic heme and a large suite of other metabolic products (42, 43). We identified a large number of phosphorylation targets in proteins associated with signaling, ion transport, and vesicle-associated transport. Our results provide a unique opportunity to establish novel targets for mosquito-control strategies and show the value of Ae. aegypti as a model system for understanding the mechanisms of water and ion homeostasis in blood-feeding arthropods.

p.12—It is not clear to this reviewer whether ‘differentially expressed proteins’ refers to a change in phosphorylation state, change in protein abundance, or a combination of both? It would be useful to better define this term to help the reader interpret the results.

The reviewer is correct that this term is confusing within the context of our study. We have changed the text to refer to ‘differentially phosphorylated proteins.’

p.12—I would suggest using ‘transport’ instead of ‘transportation’ in the text and all figures.

We agree with the Reviewer’s suggested change and have updated the manuscript accordingly.

p.12—I would suggest using ‘powered’ instead of ‘mediated’ as the latter suggests the V-ATPase directly transports Na and K, which may confuse some readers.

We agree with the Reviewer’s suggested change and have updated the manuscript accordingly.

p.12—Remove ‘channels’ after ‘antiporter’. Antiporters are a type of membrane transporter and not a channel.

We agree with the Reviewer’s suggested change and have updated the manuscript accordingly.

p.13—‘sodium/hydrogen exchanger’ is too vague. The authors are showing changes to two specific NHE/NHA proteins in Fig. 2. One of them (A0A6I8TKG3) corresponds to an ortholog of Drosophila NHA1 (AAEL011109) and the other (A0A6I8T520) corresponds to an ortholog of Drosophila NHE2 (AAEL001503) also known as Aedes NHE3 (using nomenclature of Gill). Distinguishing these two proteins is important as they are expected to be expressed on different cell types and membranes in Malpighian tubules. See the following references for more details:

A.K. Pullikuth, K. Aimanova, W. Kang'ethe, H.R. Sanders, S.S. Gill, Molecular characterization of sodium/proton exchanger 3 (NHE3) from the yellow fever vector, Aedes aegypti, J Exp Biol, 209 (2006) 3529-3544.

M.A. Xiang, P.J. Linser, D.A. Price, W.R. Harvey, Localization of two Na+- or K+-H+ antiporters, AgNHA1 and AgNHA2, in Anopheles gambiae larval Malpighian tubules and the functional expression of AgNHA2 in yeast, J Insect Physiol, 58 (2012) 570-579.

The Reviewer is correct that this nomenclature is too vague. We have updated the manuscript to reflect the particular exchanger more accurately, which is NHE3.

p.13—Figure 2 legend—Beyenbach is misspelled.

We thank the Reviewer for catching this typo. We have corrected the misspelling in the figure legend.

p.14-15—The results sections on endomembrane and TOR contain several abbreviations and proteins that the reviewer was unfamiliar with (e.g., TSC1; sestrin; GATOR2). It may be useful to include a supplemental table that spells out each abbreviation used in these sections and provides a brief description of each protein’s known function or role in another system.

We thank the Reviewer for this suggestion. We have made a supplemental table (S2 File) containing more information on all endomembrane, TOR, and cell-cell junction proteins described in the manuscript text and figures. We have updated the ‘Phosphoprotein identification’ section of the results to include the following sentence citing this table: “Phosphoproteins presented in figures and in the text are highlighted in S2 File.”

p.16-17—The authors indicate that changes in spectrin and actin were ‘of particular interest’, but it is unclear why they are of particular interest. More context or background may help.

We have added some extra context to this sentence to indicate that spectrin and actin are important for membrane stability and cell morphology. The sentence now reads, “Of particular interest, we identified changes in phosphorylation in several proteins associated with spectrin and actin, which are involved in maintenance of cell membrane stability and cell shape and therefore may cause alteration of intercellular permeability, including spectrin α and β chains, actin-5C, adducin, and protein 4.1 homolog (12, 20).”

p.18—As noted above, the paragraph explaining the rationale for the 3 time points may be more helpful in the Intro or Methods.

We have moved this paragraph to the Introduction section.

Discussion in general: More in depth interpretation of the significance of the phosphorylation on the biochemical activity of the specific protein groups would be welcomed for interpreting potential physiological consequences in the context of Malpighian tubule function. Most protein groups mentioned are discussed in general terms and repeat the presentation of results, but do not provide much in terms of physiological interpretation, especially in the context of how the tubules are thought to be functioning at the different time points measured. More speculation and generation of specific hypotheses is encouraged.

For example, on p.19, V-ATPase--- What is the significance of the dephosphorylation on V-ATPase activity? Others have shown that PKA enhances V-ATPase activity in Aedes MTs (e.g., F. Tiburcy, K.W. Beyenbach, H. Wieczorek, Protein kinase A-dependent and -independent activation of the V-ATPase in Malpighian tubules of Aedes aegypti, J Exp Biol, 216 (2013) 881-891.). Could dephosphorylation potentially explain in part the previously documented decrease in diuresis capacity observed by Esquivel et al (2016 PeerJ) 24 h PBM in Aedes albopictus?

We have updated the “V-ATPase subunits are dephosphorylated at 24 hours PBM” Discussion section to address the Reviewer’s question. This section now reads:

“V-ATPase subunits are dephosphorylated at 24 hours PBM - According to the model of MT diuresis proposed by Beyenbach, the ion gradients required to power the osmotic flux of water into the MT lumen are powered by an electrochemical gradient created by vacuolar-type ATPase (V-ATPase) (7, 12, 13). V-ATPases are evolutionary conserved multi subunit membrane proteins with two structural domains, the intracellular V1 and the membrane standing V0 domain (57). While we did not observe any differences in phosphorylation of V-ATPase subunits between unfed and 1 hr PBM mosquitoes, we observed seven residues distributed across four V1 subunits (B, E, H, F) with decreased phosphorylation at 24 hrs PBM relative to the unfed time point (Fig 2d). Interestingly, phosphorylation of V-type ATPase subunit C has previously been implicated in the assembly of V1 and V0 units in insects (58). While we did identify phosphorylated residues in subunit C in our MT datasets (S1 File), none of these phosphorylation sites had changed phosphorylation patterns over the course of our experiment. Interestingly, a previous study on V-ATPase activity in Ae. aegypti MTs concluded that while Protein Kinase A activity was necessary for V-ATPase activation, direct phosphorylation of V-ATPase subunits may not be required for V-ATPase activation (46). Therefore, V-ATPase activity in Ae. aegypti MTs may be regulated in a different fashion than the most well-known method of subunit C phosphorylation. We propose that the observed reductions in phosphorylation of V-ATPase subunits at 24 hrs PBM is responsible for the decrease in urine excretion within 24 hrs PBM that have been observed in Ae. aegypti and other mosquito species (8, 11, 41, 59). It is also possible that more rapid or transient phosphorylation of subunit C happens in Ae. aegypti MTs prior to 1 hr PBM, and we did not observe these changes in the time points investigated in this study. Experiments to determine what effects of dephosphorylation of these subunits have on V-ATPase activity, and whether these dephosphorylation events produce the same effects as changes in subunit C phosphorylation has been shown to have (58) are necessary.”

Also on p.19, for NHE-- What effects does phosphorylation have on other NHEs (activation, deactivation)? What insights does this potentially provide into exchange activity across basolateral membrane for NHE3 and apical membrane for NHA2, and how ions are transporter? 

The Reviewer poses insightful questions, which we hope we have addressed by updating the Discussion text to read:

“Na+/H+ exchangers have altered phosphorylation patterns at early and later times PBM - We observed changes in phosphorylation of two Na+/H+ exchangers (NHE3 and NHA2) by 1 hr PBM (Fig 2). NHA2 has been shown to localize to the apical membrane of SCs in Anopheles gambiae (60), while NHE3 localizes to the basolateral membrane of MT cells (61). Interestingly, An. gambiae NHA2 gene expression was shown to decrease 3 hrs PBM (62), which was hypothesized to be due to NHA2 being used to preserve hemolymph sodium content, and therefore having reduced expression during natriuresis (60, 62). We detected an increase in NHA2 phosphoprotein levels at 1 hr PBM, and a decrease in NHA2 phosphoprotein levels by 24hrs PBM (Fig. 2). Our data provides support for this hypothesis that NHA2 activity plays an important role in sodium homeostasis by likely changing activity at different times during diuresis as Na+ excretion rates change. NHE3 phospho-S770 increased 187-fold at 1 hr PBM, while phospho-S772 decreased 167-fold at this time point. We hypothesize that the shifting phosphorylation patterns of these two phosphorylation sites could be involved in the regulation of the activity of this exchanger and might represent an ‘on-off’ switch for this protein. NHE isoforms in in other eukaryotic species have been demonstrated to be activated by phosphorylation in response to growth factor signaling (63, 64). Hormone signaling plays an important role in initiating diuresis after blood feeding (8, 17-19), so it is possible that DH31-stimulated intracellular signaling pathways in MT epithelia are responsible for the phosphorylation of NHE3 we observed in our samples.”

Also, confirm whether the NHA is NHA1 or 2. 

We have confirmed that the NHA is NHA2 by searching for the gene ID on Entrez Gene to identify the gene name in Ae. aegypti (sodium/hydrogen exchanger 9B2), and then searching for synonymous gene names, which confirmed that this gene is NHA2.

In the original paper that cloned Aedes NHE3, Pullikuth et al (2006) identified predicted phosphorylation sites. How do the observed phosphorylation sites in the present study compare with the predictions?

We have added the following text to the Discussion section to address the Reviewer’s question:

“Previous work characterizing the localization and function of Ae. aegypti NHE3 identified a predicted set of phosphorylation sites, with many enriched in the intracellular C-terminus (60). Our phosphoproteome analysis revealed a similar enrichment of phosphorylated amino acid residues in the C-terminus, but surprisingly, only one of the predicted phosphorylation sites from that study (S1017) changed phosphorylation in our study. We found only one other site from Pullikuth and colleagues’ NHE3 characterization (S1099) in our dataset. The C-terminus of NHE3 appears to be a rich region of possible phosphorylation sites as demonstrated by predictions (60) and by our analysis. Future studies to validate the differential phosphorylation of S770 and S772, as well as the S1017 residue which was both predicted by Pullikuth et al. (60) and found in our dataset will be useful for understanding how the activity of this important transporter is regulated after blood feeding.”

p. 19, aquaporins—The authors should clarify that the findings are consistent with the hypothesis at the time points measured. It is still possible that gating via phosphorylation could be involved at early time points not examined in this study (e.g., < 1h PBM).

The Reviewer is correct. We have updated the Discussion text accordingly. The text now states:

“Aquaporins are dephosphorylated at the 24 hr PBM time point - We did not observe a corresponding change in AQP phosphorylation state at 1 hr PBM. We interpret this finding as support for the hypothesis that mosquito AQPs are not regulated by gating at the time points we studied as has been shown in the regulation of plant AQPs (29). This would be consistent with the finding that human AQP2 C-terminal phosphorylation does not affect its water transport (61). We acknowledge that these data do not preclude gating by phosphorylation having a more immediate effect on water transport earlier than 1 hr PBM.”

p.20, vesicle transport proteins—The authors should clarify on what is meant by ‘membrane secretion’. This is a vague term. Are the authors referring to apical, basolateral, intracellular, or intercellular membranes? Secretion of ions, water, or wastes? 

The Reviewer is correct that this term is too vague. We have updated the text to state:

“Previous research has shown that phosphorylation of this protein plays an important role in regulating ER-Golgi vesicle transport in yeast and human cells (62, 63), so our observation of extremely increased phosphorylation after blood feeding provides confidence in our conclusion that vesicle transport between organelles and to plasma membranes of MT cells increases PBM.”

How would the authors predict the changes to endosomal content sorting into lysosomes and MVBs affect tubule function? It would be interesting for the authors to discuss their findings in the context of a previous study by Bradley et al. that found ultrastructural changes in apical brush border of Aedes MTs 24 PBM:

T.J. Bradley, D.M.J. Sauerman, J.K. Nayar, Early cellular responses in the Malpighian tubules of the mosquito Aedes taeniorhynchus to infection with Dirofilaria immitis Nematoda, Journal of Parasitology, 70 (1984) 82-88.

We thank the Reviewer for this insightful question. We have added the following text to the Discussion that we believe addresses the question raised by the Reviewer:

“The alterations in phosphorylation of proteins involved in endosomal sorting to lysosomes and MVBs (Fig. 4) are interesting, as they do not directly influence solute or ion transport. However, there are several important roles that these proteins may play during diuresis. First, they may be important for sorting damaged membrane proteins to prevent them from being recycled to the cell membrane where they would affect the efficiency of water and solute secretion. Second, they may play important roles in sorting waste products taken in from the basolateral membrane to lysosomes or MVBs to be detoxified and cleared without causing damage to the MT cells. There is evidence to support the second proposed mechanism in insect MTs, as MVB accumulation and exocytosis across the apical membranes of MT cells has been demonstrated in the ground beetle, Calathus fuscipes (64). Interestingly, a previous study in a related mosquito species, Ae. taeniorhynchus determined that by 24 hrs PBM, the microvilli brush border structure had reduced when compared to unfed mosquitoes (65). We propose that this may be due to changes in membrane allocation, with brush border membranes having large amounts of microvilli to provide surface area for fluid and solute exchange, while later after blood feeding, more membrane structures may be dedicated to endosomes, lysosomes and MVBs for detoxification and secretion of waste products.”

p.20, TOR signaling—The idea of a nutrient sensor is very interesting, but how does it potentially relate to the context of MT function? More details explaining how TOR signaling would impact known physiological functions would be helpful. Is this the first study to identity a potential TOR signalling pathway in insect Malpighian tubules? More explanation of the significance of this finding is welcomed to highlight a potentially high impact finding.

We thank the Reviewer for their suggestion. We have updated the text to include more discussion of mTOR signaling in Malpighain tubules. The text now reads:

“Proteins of the TOR signaling cascade are phosphorylated through 24 hr PBM - The TOR signaling pathway is part of a cellular nutrient sensing system and involved in regulating protein synthesis (2, 70-72). Several upstream regulators of the TOR kinase, as well as the TOR kinase substrate, S6 kinase, and its substrate, ribosomal protein S6 are enriched with phosphorylation sites that are hyperphosphorylated PBM (Fig 5a). Ribosomal protein S6 phosphorylation is thought to play an important role in new protein synthesis (2, 71-73), and we observed changes in phosphorylation of four serine residues, three of which are conserved across all of the insects and vertebrates we analyzed (Fig 5c). The hyperphosphorylation of proteins of the TOR signaling cascade remained through 24 hrs PBM (Fig 5a).

We hypothesize that TOR signaling likely is activated to initiate gene expression to facilitate the expression of genes to support the need of cells for proteins to continue diuretic and detoxification activities. Previous research in An. gambiae MTs has shown increased expression of proteins with putative involvement in amino acid transport after blood feeding (62). Additionally, we observed changes phosphorylation of proton-coupled amino acid transport proteins (S1 File), which have previously been shown to regulate mTOR signaling (74). These results suggest that the cells of mosquito MTs have a nutrient sensor that involves the TOR signaling cascade and that this cascade is activated in the MTs after a blood meal. To the best of our knowledge, this study is the first to report on the mTOR signal pathway in mosquito Malpighian tubules.”

p.20-21, cell-cell junctions—The authors should include a discussion of findings from earlier studies that provided proteomic insights into how intercellular junctions, the cytoskeleton, and apical membrane are modified by diuretic hormones involved with the post-blood meal diuresis (e.g., aedeskinin). One study in particular (Miyauchi et al. 2013) provided a detailed examination of the impact of phosphorylation of adducin that would be interesting to compare with the present findings. Another study by Karas et al. should be of interest to the authors given their interest in the actin-spectrin cytoskeleton. The authors are encouraged to cite these studies and incorporate them into the discussion where relevant.

J.T. Miyauchi, P.M. Piermarini, J.D. Yang, D.M. Gilligan, K.W. Beyenbach, Roles of PKC and phospho-adducin in transepithelial fluid secretion by Malpighian tubules of the yellow fever mosquito, Tissue Barriers, 1 (2013) 1-14.

K.W. Beyenbach, S. Baumgart, K. Lau, P.M. Piermarini, S. Zhang, Signaling to the apical membrane and to the paracellular pathway: changes in the cytosolic proteome of Aedes Malpighian tubules, J Exp Biol, 212 (2009) 329-340.

K. Karas, P. Brauer, D. Petzel, Actin redistribution in mosquito Malpighian tubules after a blood meal and cyclic AMP stimulation, J Insect Physiol, 51 (2005) 1041-1054.

We thank the Reviewer for their suggestion. We have updated the Discussion text to include more of a discussion of the implications of our findings within the context of the suggested studies:

“Changes in phosphorylation of cell junction proteins could affect paracellular water flow PBM - We also observed alterations in phosphorylation of several proteins associated with cell-cell junctions at 1 hr PBM (Fig 6), indicating that the structure of these junctions may change after blood-feeding. While some phosphorylation changes occurred at 1 hr PBM, overall more dephosphorylation of cell junction proteins occurred at the 24 hr PBM time point. Previous research has demonstrated that several phosphoproteins identified in our study dynamically re-localize out of the cytoplasm of MT cells after treatment with the diuretic hormone aedeskinin-III (69). Additionally, there is evidence that actin and adducin phosphorylation changes in response to natriuretic hormone signaling (69-71). These studies suggest that phosphorylation of these proteins allow for remodeling of the actin-spectrin cytoskeleton, which may alter PC apical brush border membrane structure or paracellular transport (69, 70). Our findings mesh well with these previous studies, as we observed different phosphorylation patterns in these proteins after bloodfeeding. In particular, actin was demonstrated to re-locate to the apical brush border of PCs in response to cyclic AMP treatment relative to non-treated PCs (71). In this study, we observed dephosporylation of Actin-5C by 24 hr PBM (Fig. 6), indicating that actin filaments may be depolymerizing at this time point. This is also supported by our findings of adducin phosphorylation, as phosphorylation of adducin leads to disassociation of actin and spectrin, and we observed reduced adducin phosphorylation at 1 hr PBM and enrichment of adducin phosphorylation at 24 hrs PBM (Fig. 6). This indicates that even though much of the water from the blood meal has been excreted, the organization of cell-cell junctions remains dynamic through the first 24 hr PBM.”

p.21, conclusion—Although the present study provides a major advance in regards to the phosphoproteomics of MTs, the authors should include some caveats regarding the limitations of their study and emphasize that their findings are only relevant to the time points examined. Thus, they cannot rule out effects at earlier time points (e.g., 5 min PBM). Given that diuresis begins as soon as blood ingestion begins and post-translational changes can occur rapidly (within minutes) the authors may want to suggest future studies examine earlier time points PBM to provide further insights into relatively immediate changes in the MT phosphoproteome that may regulate the switch-like stimulation of diuresis after blood feeding (e.g., Beyenbach et al. 2009).

We thank the reviewer for their suggestion. We have added the following section to the Discussion section that we believe addresses limitations of the study as identified by both Reviewers:

“Limitations of this study - The results reported in this paper are based on single samples containing pools of 200 MTs. We acknowledge that a single sample for each time point elevates the risk of outliers affecting our data and does not allow for statistical analysis of our dataset. However, we believe our data represent a robust phosphoprotein profile of MTs at each time point, and that a pool of 200 MTs will likely minimize the chance of one or a few outlier MTs having an outsized effect on the overall data from any sample. We also acknowledge that without a proteome sample for each time point, we cannot state with certainty that alterations in phosphorylation are due solely to kinase/phosphatase activity on existing proteins, or if some of the observed changes are due to changes in synthesis/degradation of particular proteins. Future studies to validate how much of our observed changes are due to altered phosphorylation of existing proteins, and how much are due to new protein synthesis will prove useful to validate these data. These validation experiments will be performed in future studies based on these data. We chose not to sample mosquitoes from any time point prior to 1 hr PBM to ensure that we could standardize our samples. As diuresis begins within minutes of the initiation of a bloodmeal and urine production can last up to three hours PBM, we believe that selecting 1 hr PBM as our “early” time point gives an accurate profile of early protein regulation, but we acknowledge that phosphorylation events likely happen rapidly after initiation of blood feeding. This problem could be solved with the use of a blood feeding system allowing for blood feeding and sampling of individual mosquitoes. Kumar et al. recently described a 3D-printed chip system that allows for feeding and monitoring of individual mosquitoes (72), which would prove useful for future experiments to assay for rapid changes in protein phosphorylation within the first hour of blood feeding.”

Figure 1-- The illustration of the mosquito is not described in the legend. Its context is unclear based on the legend and the text in which it is cited. Eliminate the mosquito or add a description. The cartoon of the different regulatory mechanisms is very clear and effective.

We thank the Reviewer for pointing out this oversight. We have updated the Figure legend to read: 

“Figure 1. Alternative hypothetical mechanisms of transporter activation in MT cells, using AQPs as an example. Top: After blood feeding, female mosquitoes must rapidly eliminate large amounts of water and sodium from the ingested blood meal. Water, ions and nutrients from the blood meal are absorbed from the midgut into the hemolymph, and excess water and solutes are re-absorbed by the MTs and excreted in the process called diuresis. Bottom: Illustrated are three possible mechanisms of transporter activation in MT cells during diuresis. The quickest mechanism of activation is gating, or activation of transporters already inserted into the plasma membrane by post-translational modification such as phosphorylation or membrane voltage changes. A second rapid method of activation is trafficking of storage vesicles containing existing transporters to the plasma membrane. The slowest of the three mechanisms is de novo gene expression of transporter proteins in response to diuresis signaling.”

Figure 2b-- Only Na is shown to be antiported across the apical membrane in the figure. K+ should also be included to reflect what is stated in the text. Also, there is no evidence for apical K+ channels in principal cells of mosquito tubules. This should be removed from the model. There is evidence for basolateral K+ channels in principal and stellate cells (see reference below):

P.M. Piermarini, S.M. Dunemann, M.F. Rouhier, T.L. Calkins, R. Raphemot, J.S. Denton, R.M. Hine, K.W. Beyenbach, Localization and role of inward rectifier K+ channels in Malpighian tubules of the yellow fever mosquito Aedes aegypti, Insect Biochem Mol Biol, 67 (2015) 59-73.

We thank the Reviewer for correcting this error. We have updated the figure accordingly.

Table S1—It would be convenient to the reader if the gene names (column E) could all be standardized to the vectorbase ID#s (e.g., AAEL) for facilitating follow ups on orthologs and available data in vectorbase.

We agree with the Reviewer, and have updated S1 File accordingly. 

 

Reviewer #2: This MS presents an interesting and thought-provoking dataset, and will be a useful contribution to the field.

We thank the Reviewer for their favorable assessment of our manuscript.

Major points:

More detail is needed on how differential expression is scored. If, say, a protein doubles its expression level between t=1 and t=1h, but its percentage phosphorylation remains the same, would it show as the same, or 100% upregulated, in these data? Change in expression level is also a perfectly good way of regulating activity! 

The Reviewer is correct that changes in protein expression may explain some of the changes in the phosphoprotein data we report in this study. We have added the following text to the Discussion to address this point:

“We also acknowledge that without a proteome sample for each time point, we cannot state with certainty that alterations in phosphorylation are due solely to kinase/phosphatase activity on existing proteins, or if some of the observed changes are due to changes in synthesis/degradation of particular proteins. Future studies to validate how much of our observed changes are due to altered phosphorylation of existing proteins, and how much are due to new protein synthesis will prove useful to validate these data. These validation experiments will be performed in future studies based on these data.”

The broader issue of normalization of data also needs to be addressed explicitly.

The Reviewer’s point is well taken. We have updated the ‘Data Analysis’ section of the Methods to include a sentence describing how the data was normalized: “Sample protein data were normalized using the MaxLFQ package (44) in MaxQuant.”

Like most ‘big data’ papers, the results are over-stated. 

We hope that our revisions throughout the manuscript, particularly in the discussion satisfy the Reviewer’s concerns about overstating our results.

I’m particularly exercised by figs 3-6. 

We apologize for any confusion created by these figure legends, but we believe that the figures and legends do contain the requested information in their existing state. We hope the answers to the following questions about figures will satisfy the Reviewer’s concerns.

Where did these graphics come from? Are they BioRenders of human pathways? 

The graphics referred to by the Reviewer were generated by the authors in Adobe Illustrator using known information about the roles played by these proteins in vesicle transport, TOR signaling and cell-cell junctions in eukaryotic organisms. 

Are any of these proteins /pathways experimentally validated in Aedes? 

Homologs for these pathways, and many of the proteins in these pathways are conserved across eukaryotes, from yeast to humans, with the major exception of some cell-cell junction proteins that differ between invertebrates and vertebrates. In the case of these cell-junction proteins, we cited a review in the legend of Figure 6 (Beyenbach & Piermarini, 2011) that includes information on proteins associated with mosquito cell junctions that we referred to when analyzing our data. 

The legends need to be far more specific, along the lines “Human vesicle trafficking pathway (BioRender), together with Aedes homologs with significant phosphorylation differences. Proteins with upregulated phosphorylation at both time points are coloured green, down regulated red, and proteins with mixed phosphorylation are coloured red/green or as appropriate. Proteins not detected, or without significant changes, or with no Aedes homologs, are left unshaded.” Appropriately colour coding the figure would then be much more useful to the reader.

Because we are including phosphorylation data from multiple time points, we chose not to color code the proteins themselves to alleviate confusion in the figure. Rather, we included tables of proteins, with fold-changes between unfed and either 1 hr PBM or 24 hr PBM time points, and included colored arrows to indicate whether the phosphorylation state was increased or decreased at each PBM time point relative to unfed mosquitoes. We believe this is a suitable way to demonstrate the detected changes in phosphorylation.

The Reviewer’s suggestion to ensure we more clearly state that the proteins we have illustrated in our figures are Aedes homologs is well taken, and we have updated the figure legends to more clearly state that all proteins represented in the figures are Ae. aegypti proteins that are conserved in other eukaryotes. Each figure legend for Figs. 3-6 now includes a descriptive sentence similar to this (taken from Fig. 3 as an example): “Ae. aegypti proteins associated with co-translation and ER-Golgi transport conserved in other eukaryotic organisms are illustrated.”

Similarly, the authors should go through the MS and check that they qualify their claims suitably.

We believe that in addressing the comments from both Reviewers, we have qualified our claims and hopefully have satisfied the Reviewer.

Some of the changes are truly remarkable. The authors may want to bring these out into a separate table; but also carefully scrutinise the raw data to ensure that the proteins are truly expressed at all timepoints – otherwise, these massive values may reflect ‘divide by near zero’ artefacts.

We have ensured that data reported for specific phosphoproteins were expressed at all time points. We do appreciate the Reviewer’s suggestion of separating some of the most extreme changes into a separate table, and we have done this (S3 File). This will allow for identification of other phosphoproteins outside of the pathways we focused on that may be interesting targets for future experiments.

Some of the more original points need independent validation; the data presented are based on a small number of runs from a commercial provider; did the authors try to independently validate any of these phosphorylation changes themselves?

The Reviewer’s point is well taken. We have not independently validated any of these phosphorylation changes at this point. The purpose of this paper is to provide an overview of phosphorylation changes in the MTs of Ae. aegypti. We do plan to perform follow-up studies to validate some of the more interesting phosphorylation events and identify how changes in phosphorylation may affect MT function.

Minor points:

Cross-referencing against microarray data. This could be taken a little further, and would be interesting. Although the authors cite Esquivel et al., they may also want to have a look at Overend et al (https://doi.org/10.1016/j.ibmb.2015.05.007) – although this paper was in Anopheles, the underlying microarray (Affymetrix) was much more comprehensive. And very recently, there’s another Aedes paper – Hixson et al (https://doi.org/10.7554/eLife.76132), that may be of use.

We thank the Reviewer for suggesting these papers. The study by Overend et al. (2015) in particular has several interesting findings related to our study, and we have included references to them when appropriate throughout our discussion of our phosphoproteomics results.

V-ATPases: The authors suggest that multiple subunits change their phosphorylation status. This is a big deal, because despite several decades of looking, the field is really satisfied with the data only for subunit C – which these authors find to be unchanged. Rather than focus on Aedes, they should back up their claims of phosphorylation of key targets with evidence to show that this is known to happen elsewhere. The V-ATPase, incidentally, is one example where direct independent verification would be welcome.

The Reviewer’s point is well taken. We have updated the text in the Discussion to read:

“V-ATPase subunits are dephosphorylated at 24 hours PBM - According to the model of MT diuresis proposed by Beyenbach, the ion gradients required to power the osmotic flux of water into the MT lumen are powered by an electrochemical gradient created by vacuolar-type ATPase (V-ATPase) (7, 12, 13). V-ATPases are evolutionary conserved multi subunit membrane proteins with two structural domains, the intracellular V1 and the membrane standing V0 domain (57). While we did not observe any differences in phosphorylation of V-ATPase subunits between unfed and 1 hr PBM mosquitoes, we observed seven residues distributed across four V1 subunits (B, E, H, F) with decreased phosphorylation at 24 hrs PBM relative to the unfed time point (Fig 2d). Interestingly, phosphorylation of V-type ATPase subunit C has previously been implicated in the assembly of V1 and V0 units in insects (58). While we did identify phosphorylated residues in subunit C in our MT datasets (S1 File), none of these phosphorylation sites had changed phosphorylation patterns over the course of our experiment. Interestingly, a previous study on V-ATPase activity in Ae. aegypti MTs concluded that while Protein Kinase A activity was necessary for V-ATPase activation, direct phosphorylation of V-ATPase subunits may not be required for V-ATPase activation (46). Therefore, V-ATPase activity in Ae. aegypti MTs may be regulated in a different fashion than the most well-known method of subunit C phosphorylation. We propose that the observed reductions in phosphorylation of V-ATPase subunits at 24 hrs PBM is responsible for the decrease in urine excretion within 24 hrs PBM that have been observed in Ae. aegypti and other mosquito species (8, 11, 41, 59). It is also possible that more rapid or transient phosphorylation of subunit C happens in Ae. aegypti MTs prior to 1 hr PBM, and we did not observe these changes in the time points investigated in this study. Experiments to determine what effects of dephosphorylation of these subunits have on V-ATPase activity, and whether these dephosphorylation events produce the same effects as changes in subunit C phosphorylation has been shown to have (58) are necessary.”

Statistics - I didn't notice it, but was Bonferroni or other correction made for multiple data?

We apologize for any confusion, but no statistical analysis was performed. We have updated the “Mosquito Rearing and Sampling” portion of the Methods section to make this point more clear. The text now reads: 

“One hr and 24hr post blood meal mosquitoes were anesthetized on ice and 200 Malpighian Tubules (MT) were dissected in modified Aedes physiological saline (mAPS) solution from each group. 200 MTs were dissected from unfed mosquitoes as a control. Dissected All 200 dissected MTs from each time point were pooled to generate samples, and samples were frozen in 200µL mAPS containing 2µL HaltTM protease inhibitor cocktail (Thermo Scientific, Rockford, IL) and 2µL HaltTM phosphatase inhibitor cocktail (Thermo Scientific, Rockford, IL).” 

We have also added the following text to the discussion to address this as well:

“Limitations of this study - The results reported in this paper are based on single samples containing pools of 200 MTs. We acknowledge that a single sample for each time point elevates the risk of outliers affecting our data and does not allow for statistical analysis of our dataset. However, we believe our data represent a robust phosphoprotein profile of MTs at each time point, and that a pool of 200 MTs will likely minimize the chance of one or a few outlier MTs having an outsized effect on the overall data from any sample.”

---

## [Decision Letter · Decision Letter 1]

20 Jun 2022

PONE-D-22-09152R1Phosphoproteomics profiling of Aedes aegypti Malpighian tubules during blood meal processing reveals dramatic transition in functionPLOS ONE

Dear Dr. Hansen,

Thank you for submitting your manuscript to PLOS ONE. After careful consideration, we feel that it has merit but does not fully meet PLOS ONE’s publication criteria as it currently stands. Therefore, we invite you to submit a revised version of the manuscript that addresses the points raised during the review process.

This is a somewhat uncommon as the reviewers had opposite conclusions in the second round. Reviewer #2 now changed his evaluation to rejection. He realized from your reply that the paper is based on a single experiment without biological replicates, which is a fact that I also had not acknowledged in the first round. As you can see from his comment, he thinks that without proper replication, the results should not be published. I agree that this is a significant drawback in the experimental design, but I think the results can still help the field go forward. My point here is that when results highlight several proteins belonging to the same pathways or known to be physiologically connected, this increase the probability of identifying an observation with real biological significance, despite the lack of sampling statistics. Although you have introduced one mention to this limitation in the  discussion, this should be mentioned also in the methods section. However, this limitation calls for a stronger emphasis that this is an exploratory work (this should be clear from the beginning) whose role is to generate hypothesis at the end of the paper and provide evidence that may re-orient other people´s work.

We look forward to receiving your revised manuscript.

Kind regards,

Pedro L. Oliveira

Academic Editor

PLOS ONE

Reviewers' comments:

Reviewer's Responses to Questions

**Comments to the Author**

1. If the authors have adequately addressed your comments raised in a previous round of review and you feel that this manuscript is now acceptable for publication, you may indicate that here to bypass the “Comments to the Author” section, enter your conflict of interest statement in the “Confidential to Editor” section, and submit your "Accept" recommendation.

Reviewer #1: All comments have been addressed

Reviewer #2: (No Response)

2. Is the manuscript technically sound, and do the data support the conclusions?

Reviewer #1: Yes

Reviewer #2: No

3. Has the statistical analysis been performed appropriately and rigorously? 

Reviewer #1: Yes

Reviewer #2: No

4. Have the authors made all data underlying the findings in their manuscript fully available?

Reviewer #1: (No Response)

Reviewer #2: Yes

5. Is the manuscript presented in an intelligible fashion and written in standard English?

Reviewer #1: Yes

Reviewer #2: Yes

6. Review Comments to the Author

Reviewer #1: Thanks to the authors for addressing the suggested changes and responding to all of the inquiries. The authors revisions have strengthened an already strong submission. The hypothesized physiological implications of the phosphorylation changes in the context of MT function are now well explained. Congrats on a fine study and manuscript.

Just a few more minor edits in the revision remain below:

p.18 of revision: replace 'intercellular' with 'paracellular'. The proteins may mediate connections between cells, but the effect on permeablity would be paracellular. Intercellular permeability would refer to connections between cells (e.g., gap junctions).

p.21: "has been shown to have" is awkward in the context used. I think a word or two is missing. Edit this phrase

Reviewer #2: The authors have provided a detailed reply to the reviewers’ questions; but I’m afraid that as a result, I can no longer support publication.

The authors are reporting on changes reported in single samples, without any statistical analysis, let alone multiple sampling correction. Although changes have been seen, I’m afraid nothing can be asserted from these data. The experimental design should have incorporated multiple (4+) biological replicates of each sample point, allowing statistically significant changes to be identified. Without this rigor, the dataset is not publishable.

7. PLOS authors have the option to publish the peer review history of their article (what does this mean?). If published, this will include your full peer review and any attached files.

Reviewer #1: No

Reviewer #2: No

---

## [Author Response · Author response to Decision Letter 1]

22 Jun 2022

Thank you for your honest assessment of our manuscript. We have addressed your specific concerns below and have also addressed the Reviewers’ comments in detail. Changes in the text from the previous version to this one are emphasized in bold in all responses below.

My point here is that when results highlight several proteins belonging to the same pathways or known to be physiologically connected, this increase the probability of identifying an observation with real biological significance, despite the lack of sampling statistics. 

We agree with you that these data likely do include biologically significant results despite there being only single biological replicates for each time point. We have included the following text in the “Limitations of this study” section of the Discussion to address this point. The text now reads: 

“Limitations of this study - The results reported in this paper are based on single samples containing pools of MTs from 200 mosquitoes. We acknowledge that a single sample for each time point elevates the risk of outliers affecting our data and does not allow for statistical analysis of our dataset. However, we believe our data represent a robust phosphoprotein profile of MTs at each time point, and that pools of MTs from 200 individuals will likely minimize the chance of one or a few outlier MTs having an outsized effect on the overall data from any sample. Additionally, we reported on pathways and processes with many associated proteins that underwent changes in phosphorylation post-blood meal, and therefore the likelihood that we identified changes in phosphoproteins with real biological significance is high, despite the lack of sample size.”

Although you have introduced one mention to this limitation in the discussion, this should be mentioned also in the methods section. However, this limitation calls for a stronger emphasis that this is an exploratory work (this should be clear from the beginning) whose role is to generate hypothesis at the end of the paper and provide evidence that may re-orient other people´s work.

We agree that more emphasis could be placed on this limitation and have updated the text of the manuscript in several places to address this.

1. We have updated the title to emphasize this point. Our new title is:

Exploratory phosphoproteomics profiling of Aedes aegypti Malpighian tubules during blood meal processing reveals dramatic transition in function.

2. We have updated the abstract to emphasize this point early on in the article. The abstract now reads: 

“Malpighian tubules, the renal organs of mosquitoes, facilitate the rapid dehydration of blood meals through aquaporin-mediated osmosis. We performed phosphoproteomics analysis of three Malpighian tubule protein-libraries (1000 tubules/sample) from unfed female mosquitoes as well as one and 24 hours after a blood meal. We identified 4663 putative phosphorylation sites in 1955 different proteins. Our exploratory dataset reveals bloodmeal-induced changes in phosphorylation patterns in many subunits of V-ATPase, proteins of the target of rapamycin signaling pathway, vesicle-mediated protein transport proteins, proteins involved in monocarboxylate transport, and aquaporins. Our phosphoproteomics data suggest the involvement of a variety of new pathways including nutrient-signaling, membrane protein shuttling, and paracellular water flow in the regulation of urine excretion. Our results support a model in which aquaporin channels translocate from intracellular vesicles to the cell membrane of stellate cells and the brush border membrane of principal cells upon blood feeding.”

3. The final paragraph of the introduction now states:

“The goal of this study was to improve our understanding of the mechanisms that regulate ions and water balance in mosquitoes after a blood meal. We investigated changes in MT protein phosphorylation by pooling MTs from 200 mosquitoes to generate single samples of 1000 tubules each, at three time points before and after blood feeding. The three time points we chose for our phosphoproteomics analysis cover the MTs in three very different states. The initial time point included unfed mosquito females that were approximately 72 hours after hatching and in a reproductive state-of-arrest (40). Little diuresis is happening during this stage (41). The second time point, one hour after a blood meal, covers the MT at their peak of activity during the rapid dehydration of the blood meal (6). At the third time point 24 hours PBM, the initial dehydration of the blood meal is completed but digestion of the blood proteins, hemoglobin and serum proteins, is in full swing producing toxic heme and a large suite of other metabolic products (42, 43). We identified a large number of phosphorylation targets in proteins associated with signaling, ion transport, and vesicle-associated transport. The results of our exploratory study provide a unique opportunity for the development of hypotheses to identify novel targets for mosquito control strategies and show the value of Ae. aegypti as a model system for understanding the mechanisms of water and ion homeostasis in blood-feeding arthropods.”

4. The second paragraph of the “Mosquito Rearing and Sampling” section of the Methods now reads:

“One week PE, adult Ae. aegypti females were fed on defibrinated bovine blood (HemoStat Laboratories, Dixon, CA) for five min. (1 hr PBM samples) or 30 min. (24 hr PBM samples). One hr and 24hr post blood meal mosquitoes were anesthetized on ice and Malpighian Tubules (MT) from 200 mosquitoes (total of 1000 tubules per sample) were dissected in modified Aedes physiological saline (mAPS) solution from each group. Additionally, MTs were dissected from 200 unfed mosquitoes as a control. All dissected MTs from each time point were pooled to generate single samples, which were frozen in 200µL mAPS containing 2µL HaltTM protease inhibitor cocktail (Thermo Scientific, Rockford, IL) and 2µL HaltTM phosphatase inhibitor cocktail (Thermo Scientific, Rockford, IL). Frozen samples were shipped on dry ice to Creative Proteomics (Shirley, New York), who performed protein extractions, phospho-peptide enrichment, and phosphoproteomics analysis.”

5. We have altered the text in the Discussion and Conclusion sections to emphasize that this study is exploratory in nature and aimed at providing data useful for generation of novel hypotheses.

a. The first paragraph of the discussion has been updated to read:

“The Malpighian tubules of insects were named in honor of the 17th century Italian scientist Marcello Malpighi who discovered pulmonary capillaries and alveoli in vertebrates as well as spiracles in insects (53). Studied for centuries, their development, anatomy, and function was thought to be well understood (54). However, using new -omics approaches, researchers have recently identified a large number of genes/proteins expressed in the MTs that suggest novel functions in metabolism, detoxification, immunity, and other cellular and organismal processes (13). Our present exploratory study adds new information by identifying thousands of phosphopeptides that are present in the Malpighian tubules of mosquitoes and by showing changes in their phosphorylation patterns after a blood meal. Phosphorylation of proteins is a posttranslational modification that has been shown to affect protein stability, protein/protein interactions, and protein function (55, 56). Kinases and phosphatases regulate basically every aspect of cellular life (57). Therefore, phosphoproteomics analysis of mosquito Malpighian tubules does not only provide an opportunity to develop hypothesis of how known processes are regulated within the cells of the tubules but also to identify new ones.”

b. The Conclusion section now reads:

“Taken together, our phosphoproteome analysis provides evidence that supports the model of MT activity proposed by Beyenbach and colleagues. We suggest that AQP activity in insect MT is not regulated by gating but rather shuttling and changes in gene expression. The phosphoproteome data generated in this exploratory study both represent a preliminary picture of the processes that occur in female Ae. aegypti after a blood meal. They provide a first baseline for the development of hypotheses and future studies that will involve pharmaceutical and molecular manipulations of individual proteins.”

We have also made several other minor changes that we believe will add clarity at certain points of the manuscript:

1. We have also updated the text throughout to clarify that our samples are pools of Malpighian tubules from 200 mosquitoes, and not pools of 200 tubules (which would only require sampling 40 mosquitoes). We have also updated the introduction text to clarify the point that Ae. aegypti have five tubules. The text now reads as follows:

“Malpighian Tubules (MTs) are the primary excretory tissue in almost all insects including Ae. aegypti, and they are functionally analogous to vertebrate kidneys (12, 13). MTs in insects are clusters of tubular organs that open into the insect hindgut. The number of tubules varies across different insect taxa, but in Ae. aegypti, this excretory organ consists of a cluster of five tubules. Each tubule is composed of two major cell types, principal cells (PC) and stellate cells (SC).”

2. We have also re-worded the following sentence in the Discussion section to remove confusion over whether findings from a different insect provide “evidence” that MVB formation is happening in Ae. aegypti MTs. The sentence now reads:

“Research on the ground beetle, Calathus fuscipes has shown that MVB accumulation and exocytosis occurs in MTs of this insect (69), which provides support for our second proposed mechanism.”

In addition to these changes, and those made at the recommendation of Reviewer 1 (see below), we have made several small changes to correct typos and grammatical errors.

We look forward to receiving your revised manuscript.

Kind regards,

Pedro L. Oliveira

Academic Editor

PLOS ONE

 

Detailed Response to Reviewers

Reviewers' comments:

Reviewer #1: Thanks to the authors for addressing the suggested changes and responding to all of the inquiries. The authors revisions have strengthened an already strong submission. The hypothesized physiological implications of the phosphorylation changes in the context of MT function are now well explained. Congrats on a fine study and manuscript.

We thank the Reviewer for their thorough critiquing of our manuscript over the course of the review process, and for their favorable assessment of our work.

Just a few more minor edits in the revision remain below:

p.18 of revision: replace 'intercellular' with 'paracellular'. The proteins may mediate connections between cells, but the effect on permeablity would be paracellular. Intercellular permeability would refer to connections between cells (e.g., gap junctions).

We thank the Reviewer for this suggestion. We have updated the text accordingly.

p.21: "has been shown to have" is awkward in the context used. I think a word or two is missing. Edit this phrase

We thank the Reviewer for this suggestion and agree that the language was awkward. We have updated the text to read:

“Experiments to determine what effects of dephosphorylation of these subunits have on V-ATPase activity, and whether these dephosphorylation events produce the same effects as those seen after changes in subunit C phosphorylation (59) are necessary.”

Reviewer #2: The authors have provided a detailed reply to the reviewers’ questions; but I’m afraid that as a result, I can no longer support publication.

The authors are reporting on changes reported in single samples, without any statistical analysis, let alone multiple sampling correction. Although changes have been seen, I’m afraid nothing can be asserted from these data. The experimental design should have incorporated multiple (4+) biological replicates of each sample point, allowing statistically significant changes to be identified. Without this rigor, the dataset is not publishable.

We thank the Reviewer for their honest assessment of our manuscript. We understand the Reviewer’s concerns, and we hope that they may be somewhat alleviated by the increased emphasis throughout the manuscript that this research project was exploratory in nature and the data are intended for presentation to the scientific community to inform the formation of new hypotheses and experimental design.

---

## [Editor Report · Decision Letter 2]

27 Jun 2022

Exploratory phosphoproteomics profiling of Aedes aegypti Malpighian tubules during blood meal processing reveals dramatic transition in function

PONE-D-22-09152R2

Dear Dr. Hansen,

We’re pleased to inform you that your manuscript has been judged scientifically suitable for publication and will be formally accepted for publication once it meets all outstanding technical requirements.

Kind regards,

Pedro L. Oliveira

Academic Editor

PLOS ONE
---

## [Editor Report · Acceptance letter]

28 Jun 2022

PONE-D-22-09152R2 

Exploratory phosphoproteomics profiling of *Aedes aegypti* Malpighian tubules during blood meal processing reveals dramatic transition in function 

Dear Dr. Hansen:

I'm pleased to inform you that your manuscript has been deemed suitable for publication in PLOS ONE. Congratulations! Your manuscript is now with our production department. 

Kind regards, 

on behalf of

Dr. Pedro L. Oliveira 

Academic Editor

PLOS ONE